# Secure 3D data hiding through cryptographic steganalysis resistance: reducing geometric inconsistency vulnerabilities

Muhammad Sajid[1], Kaleem Razzaq Malik[1], Sohail Jabbar[2], Umar Raza[3] and Muhammad Asif Habib[2]

[1] Department of Computer Science, Air University, Islamabad, Capital, Pakistan
[2] Department of Computer Science, College of Computer and Information Sciences, Imam Mohammad Ibn Saud Islamic University (IMSIU), Riyadh, Saudi Arabia
[3] Department of Engineering, Manchester Metropolitan University, Manchester, United Kingdom



Corresponding author
Sohail Jabbar, sjjabar@imamu.edu.sa

## ABSTRACT

The advent of the metaverse has generated considerable interest in 3D models, although data transfer security continues to be a paramount issue. In the contemporary digital landscape, characterized by ubiquitous internet connectivity and widespread image distribution, the protection of sensitive data within 3D models is becoming increasingly imperative. Protecting private and sensitive information within 3D models has become essential in the current interconnected digital environment, which is marked by pervasive internet access and extensive model sharing. Existing transmission mechanisms are vulnerable to various cyber risks during the transfer of important 3D models *via* insecure networks. To address the challenges in securing sensitive information embedded in 3D models, this article introduces a contemporary and effective system that combines cryptography with 3D steganography techniques. This study employed AES-128 with cipher block chaining (CBC-IV) and an initialization vector to convert plaintext into ciphertext. The study employed SHA-256, salt, and a 32-bit password to produce the encryption key, creating a fundamental layer of protection. This research used encrypted data within a 3D facial model employing geometric characteristics. This study defined key regions, identified significant vertices, and assessed the importance of each vertex based on geometric characteristics. The present study included data on vertices adjacent to landmarks, which were rounded and augmented using an enlarged scale factor, resulting in a stego 3D model. The performance measurements show how well our method works, with a Peak Signal-to-Noise Ratio (PSNR) of 61.31 dB, a Mean Square Error (MSE) of 3.17, a correlation coefficient of 0.95, and a Region Hausdorff Distance (RHD) of 0.04. Our method attained Number of Pixel Change Rate (NPCR) and Unified Average Changing Intensity (UACI) values of 94.82 and 28.31, respectively, surpassing current methodologies. Our methodology addresses geometric inconsistency issues and adeptly conceals the model's deformed geometry. In the future, we will investigate blockchain technology alongside 3D model encryption to enhance the security, authenticity, and transparency of safeguarded 3D model data.

## INTRODUCTION

The widespread adoption of 3D models as a dominant data format, propelled by improvements in computer graphics, virtual reality (VR), augmented reality (AR), and notably the metaverse, has inherently broadened the domain of steganography into an additional dimension (*Kheddar et al., 2024*). 3D steganography started to gain prominence as academics investigated the distinctive characteristics of 3D models for data embedding. In contrast to 2D images, 3D models exhibit intricate geometric configurations, comprising vertices, edges, faces, and topological relationships, hence providing a more enriched embedding space. Initial endeavors frequently employed techniques from 2D steganography (*Darani et al., 2024*). The fundamental concept of 3D steganography is concealing confidential information within a 3D model while ensuring that the model's visual aesthetics and structural integrity remain largely unaltered. This entails altering multiple attributes of the 3D model. Geometric domain methods are the predominant approach. They entail altering the geometric attributes of the model, like vertex coordinates, normal vectors, or mesh connectivity (*Sharma et al., 2024*). Topological Domain Methods utilize the connectivity links among vertices, edges, and faces. They tend to be more intricate yet can provide more resilience. The increasing need for 3D steganography arises from its distinctive benefits and the intrinsic drawbacks of other data protection techniques concerning 3D models.

To safeguard essential information included inside 3D models, numerous advantages are there, notably communication and secrecy. Another advantage is the augmented security layer; when integrated with cryptography, steganography provides an additional protection mechanism. Another benefit is that the complex and multi-faceted nature of 3D model data renders it intrinsically more challenging to evaluate for concealed information than simpler media forms (*Zhang & Hu, 2024*). Due to inadvertent embedding, there can be discrepancies that result in geometric features, like voids, self-intersections, or non-manifold edges in the stego model, rendering it visually conspicuous or impractical. Current transmission systems are susceptible to numerous cyber threats when transferring critical 3D models across insecure networks (*Long et al., 2024*). This requires an advanced strategy for safeguarding sensitive data (*SaberiKamarposhti, Ghorbani & Yadollahi, 2024*) in 3D models, assuring both its secrecy and its subtle, resilient concealment against emerging attacks. This pressing necessity propels research into sophisticated systems that adeptly integrate cryptographic robustness with smart 3D steganography methodologies.

In contemporary society, advancements in technology, especially in networked communication, have made trust in the privacy of information during transmissions an increasingly vital issue (*Sajid et al., 2025a*). The use of these new technologies facilitates the efficient transmission and storage of digital images across various networks and devices due to their simplicity, energy efficiency, and informational attributes (*Ahmad et al., 2024*).

Digital images, being a prevalent data format, frequently include significant personal and confidential information (*SaberiKamarposhti, Ghorbani & Yadollahi, 2024*). Digital images, in contrast to conventional textual information, exhibit attributes such as robust inter-pixel correlations, substantial data capacities, and significant redundancies (*Hadj Brahim, Ali Pacha & Hadj Said, 2024*; *Alexan, Alexan & Gabr, 2023*; *Sajid et al., 2025b*). In addition to personal information security concerns, similar obstacles also arise in the sectors of industry, healthcare (*Sajid et al., 2022, 2025c*), and commerce. (*Kocak et al., 2024*; *Biban, Chugh & Panwar, 2023*; *Hadj Brahim, Ali Pacha & Hadj Said, 2023*). To avert the revelation of data linked to images, methodologies such as information hiding (*Hao et al., 2023*; *Sun, Liu & Zhang, 2023*; *Sajid et al., 2024*), watermarking (*Xiao et al., 2020*), and image encryption (*Alexan et al., 2024*; *Hasan et al., 2024*; *Awais et al., 2024*) have been proposed in the existing studies.

The algorithms used in cryptography (*Buchmann, 2004*; *Wen, Lin & Feng, 2024*; *Hao et al., 2023*) are categorized into three types: symmetrical encryption (*Alshammari et al., 2021*), asymmetrical encryption (*Du & Ye, 2023*), and a combination of both (*Ettiyan & Geetha, 2023*). The process of decryption involves the utilization of the same secret key to reverse the operations that were carried out on the ciphertext, thereby recovering the original image (*Gupta & Chauhan, 2024*; *Zheng & Hu, 2021*; *SaberiKamarposhti, Ghorbani & Yadollahi, 2024*). As the number of threats to encrypted data has increased, researchers have been motivated to develop encryption methods that are both secure and more efficient (*Adeniyi et al., 2022*; *Usmonov, 2024*; *Cobb & Macoubrie, 2020*). This study has employed AES-128 to convert plaintext into ciphertext. The proposed architecture can encrypt and decode sensitive data, thereby safeguarding private information while it is being transmitted across channels of communication that are not secure (*Malik et al., 2020*). Because of this, the possibility of being intercepted or observed without authorization is reduced (*Mou & Dong, 2024*).

In the past several years, the rapid advancement of 3D technology has led to the widespread utilization of 3D models in a variety of applications, such as virtual reality and 3D printing (*Ren, Feng & Chen, 2023*; *Sajid et al., 2025d*). The protection of 3D model data has become necessary as a result of the widespread utilization of digital multimedia content sharing on the internet (*Li et al., 2024*; *Zhang et al., 2024*; *Gabr et al., 2024*). Meshes are the representations that 3D steganographers choose among these representations because of their higher capability for message embedding (*Nigro et al., 2024*). The complexity of 3D models makes it difficult to conceal information because of their complexity. For instance, it is difficult to differentiate between the more significant components of the model and the less significant components while maintaining the highest possible level of visual quality. Several algorithms, such as *Wang & Lo (2024)* and *Guan et al. (2024)*, have been developed to conceal information in 3D models. These algorithms are designed to be robust and unobtrusive to protect the data contained within these models from any potential dangers that may arise. Compared to two-dimensional graphics, 3D models provide a more genuine experience and have a more significant impact on the viewer's eyes. One of the most difficult challenges is to ensure that the transfer of 3D models is done securely. This is even though several 3D model encryption methods have been proposed

(*Xu, Zhao & Mou, 2020*; *Sun, 2021*; *Wang, Xu & Li, 2019*); nevertheless, these methods have several shortcomings.

Assuring security, safety, high capacity, and imperceptibility, this study proposes a multi-layered, secure, and dependable technique to solve the constraints and issues of insufficient security for sensitive data transferred over insecure communication channels. The methodology is designed to address the implications of insufficient security. This research utilized a combination of the Advanced Encryption Standard (AES) encryption method in cryptography and salient features-based 3D steganography to safely insert messages within 3D models, which resulted in the creation of a stego 3D model. The purpose of this research was to produce a cryptographically secure key by combining the process of creating a secure password with the SHA-256 hashing algorithm to improve the randomness and strength of the key. Within the scope of this study, landmark regions were identified, landmark vertices were determined, and the significance of each vertex was assessed based on specific geometric characteristics. The result was a 3D stego model, achieved by integrating data into vertices adjacent to landmarks through the use of rounding and an enlarged scale factor. Through the use of various protection levels, this study improves data security, privacy, and resilience against efforts to identify and extract messages that have been concealed. The primary novel contributions in tackling security concerns are as follows:

- Novel hybrid 3D model security: The research presents a modern and efficient hybrid security solution that integrates cryptography with 3D steganography techniques to safeguard sensitive information embedded in 3D models.
- Advanced encryption key generation: The research implements a rigorous approach for encryption key generation, including SHA-256 with salt and a 32-bit password. This establishes a foundational layer of defense, augmenting the security of the cryptographic element.
- Geometric-based 3D steganography: A notable advancement is the application of geometric features within a 3D facial model for data obfuscation. This entails delineating critical locations, pinpointing notable vertices, and evaluating vertex significance based on geometric attributes, particularly incorporating and enhancing vertices next to landmarks.
- Addressing geometric inconsistency: The technology is specifically engineered to rectify geometric inconsistencies and effectively obscure the model's distorted geometry resulting from the steganographic procedure. This is a significant enhancement over current methodologies.

## Organization

We structure the current research as follows: Related Work comprehensively analyzes the background. The section Methodology defines the research technique employed in this study. Embedding outlines the embedding and extraction process. 3D Seganography outlines 3D steganography and 3D model information. Evaluation Metrics outlines performance evaluation metrics. Results and Discussion delineates the findings and

interacts in discourse over these outcomes. Section Conclusion synthesizes the study's principal implications.

## RELATED WORK

This study analyzed the application of 3D steganography techniques across several domains based on their approaches.

### Practical applications and real-world examples

3D steganography can safeguard medical patient data in 3D scans by securely embedding sensitive medical information directly within the patient's MRI or CT scan data (*Rempe et al., 2025*). This guarantees that the data accompanies the image while staying undetectable to the unaided eye (*Chowa et al., 2025*). The 3D image must maintain diagnostic accuracy, free from any obvious distortions caused by concealed data that could mislead physicians (*Efe, 2025*). 3D steganography is utilized in digital watermarking (*Zhu, Wang & Gu, 2025*) to safeguard the copyright of 3D models, wherein a distinctive digital watermark is inserted into the original 3D structure. This watermark would be inconspicuous yet verifiable, assisting in establishing ownership and monitoring unwanted usage (*Mahajan & Powell, 2025*; *Alhammad et al., 2024*). Should a pirated version of the 3D model surface online, the embedded watermark would probably endure typical alterations, enabling the developer to retrieve it and substantiate their ownership. Covert communication (*Al-Ahmadi et al., 2024*) is also applicable in 3D gaming environments, where intelligence agencies or military units may utilize this technique to embed clandestine messages or coordinates within ostensibly benign 3D objects or landscapes in a shared virtual setting. Instead of employing conventional encrypted communication lines that may be subject to surveillance, the critical information is concealed directly within the game's 3D representations (*Iqbal, Khan & Lee, 2024*). The increased use of additive manufacturing (AM) and the willingness to outsource AM created a situation where sharing digital designs among various parties became normal (*Dolgavin, Yampolskiy & Yung, 2024*). Recent findings indicate that STL design files, predominantly utilized in additive manufacturing, harbor steganographic channels. Such channels enable the incorporation of further data within the STL files without altering the printed model (*Lee & Chun, 2025*). These features pose a risk of exploiting the design files as a clandestine communication medium to either exfiltrate sensitive digital data from enterprises or introduce malicious software into a secure environment.

### From text to visual content: encryption challenges in 3D models

This study conducted a comprehensive analysis to evaluate several methodologies in terms of capacity, computational cost, and robustness. Traditional encryption methods, such as Data Encryption Standard (DES) (*Dooley, 2013*) and Advanced Encryption Standard (AES) (*Daemen & Rijmen, 1999*), were initially developed to protect text files. As the number of 3D models continues to increase, there has been an increase in the demand for particular encryption methods capable of safeguarding visual content. In a study (*Joel et al., 2020*), the authors provided three symmetric algorithms, namely AES, Blowfish, and

Twofish, to demonstrate the security and performance of each method when applied to a cloud platform. Based on data processing and storage capacity, the writers concluded that each method worked uniquely. As the use of digital communications becomes more widespread, the article (*Sood & Kaur, 2023*) emphasizes the growing importance of protecting sensitive information against accidental disclosure or misuse. A detailed investigation of spatial-domain steganography is presented by the authors in a separate study (*Alhomoud, 2021*). The AES algorithm is the most important component of the image encryption approaches that are covered in the study (*Saeed & Sadiq, 2023*). During the preliminary phase, the authors encrypt the initial image by employing the AES approach. To present a novel method for generative image steganography, the authors of a study (*Sun, Liu & Zhang, 2023*) make use of guiding cues in image synthesis. The study (*Dumre & Dave, 2021*) examines the application of the least significant bit (LSB) method and the Advanced Encryption Standard (AES-128) encryption technique in the field of image steganography. An upgraded LSB approach for color image steganography was provided by the authors of the article (*Singh & Singh, 2015*). This method was designed to address issues regarding decreased data security, degraded image quality, and constrained hiding capability. The authors of the study (*Yadahalli, Rege & Sonkusare, 2020*) emphasize the utilization of two techniques for the implementation of steganography. These techniques are the discrete wavelet transform and the least significant bit approach.

## 3D model steganography: hierarchical encryption and mesh-based approaches

By implementing hierarchical decryption, the authors have recently presented a novel encryption approach (*Van Rensburg, Puech & Pedeboy, 2022*) for 3D models. Once the encryption process is complete, this approach allows for the application of a wide range of visual effects. In this process, the authors selected specific bits to create three distinct blocks, each of which undergoes independent encryption. Before the creation of the vertex element interval approach, the authors of *Ren, Ren & Zhang (2023)*, *Mukherjee, Sarkar & Mukhopadhyay (2022)* investigated a shifting strategy to successfully incorporate hidden data into a 3D image. The authors utilized a truncated space to reduce the amount of distortion and enhance the imperceptibility of the image while still preserving a significant embedding capability. An alternate technique that makes use of Parallel Breadth-First Search (PBFS) with hyper-objects was proposed by the authors of the article (*Bandyopadhyay et al., 2024*). They can insert confidential information into the points of 3D mesh images by utilizing PBFS and layer synchronization. Their methodology, on the other hand, does not provide any evidence of assault resilience. Zhang et al. proposed, in an unpublished article in 2023, a novel and customizable 3D mesh steganography system. The authors created a highly adaptive 3D mesh steganography system that prioritizes security. The technique makes use of feature-preserving distortion (FPD) to evaluate the distortion. The objective of attacks that target 3D model steganography is to determine whether or not the 3D models include any concealed information (*Girdhar & Kumar, 2018*).

**Table 1 Comparison of 3D steganography techniques.**

| Ref | Technique | Advantages | Disadvantages | Attack |
|---|---|---|---|---|
| Alkhamese et al. (2024) | Gray code | High capacity | Processing time | Noise, reordering |
| Zhang et al. (2024) | Multilayer mesh | High capacity | – | Geometric distortion |
| Bandyopadhyay et al. (2024) | Parallel BFS | High security | – | – |
| Girdhar & Kumar (2019) | BFS | Mesh traversing | Low capacity | – |
| Jiang et al. (2017) | Mapping vertex & LSB | High capacity | Low AER | Vertex reordering |
| Zhou et al. (2018) | Adaptive mesh | Strong security | Complex | – |
| Peng et al. (2025) | MeshPAD, LSB | Reduces geometric distortion | Computational complexity | Steganalysis resistance |
| Dhawan et al. (2024) | LSB, PVD, and EMD | High visual quality | Computational overhead | Multi-Layer defense |

## From conventional methods to multilayered security: a proposed framework

In comparison to the approaches that are based on compressive detection technology, conventional methods, which are presented in Table 1, necessitate a longer amount of processing time. A significant number of research studies that have been mentioned concentrate primarily on a dynamic method (Mukherjee, Sarkar & Mukhopadhyay, 2022). A gray code is utilized by the authors of another study (Alkhamese et al., 2024) to determine the vertices of 3D models that are used for embedding confidential information. An alternative method that makes use of Parallel Breadth-First Search (PBFS) with hyper-objects was proposed by the authors of the article (Bandyopadhyay et al., 2024). To solve the constraints and problems that are associated with inadequate protection for sensitive data that is transferred across insecure communication channels, this study proposes a multilayered, secure, and dependable technique. This methodology offers security, safety, high capacity, and imperceptibility.

## Relevance of proposed 3D steganography in comparison to existing methods

The suggested approach mitigates essential shortcomings present in existing 3D model encryption techniques. Although current methodologies, such as Alkhamese's Gray Code (Alkhamese et al., 2024), Zhang's multilayer mesh strategy (Zhang et al., 2024), and Bandy's Parallel BFS technique (Bandyopadhyay et al., 2024), provide distinct advantages, they are hindered by significant drawbacks such as prolonged processing durations, geometric distortion challenges, or restricted capacity. The research indicates that conventional encryption techniques, such as DES and AES, were originally developed for text data and encounter difficulties when utilized for visual content, especially 3D models. The innovation of the proposed method is in its multilayered security architecture, which integrates AES-128 encryption with cipher block chaining (CBC-IV) and an initialization vector as a fundamental layer, succeeded by advanced 3D steganography algorithms. This signifies a substantial divergence from traditional single-layer methodologies.

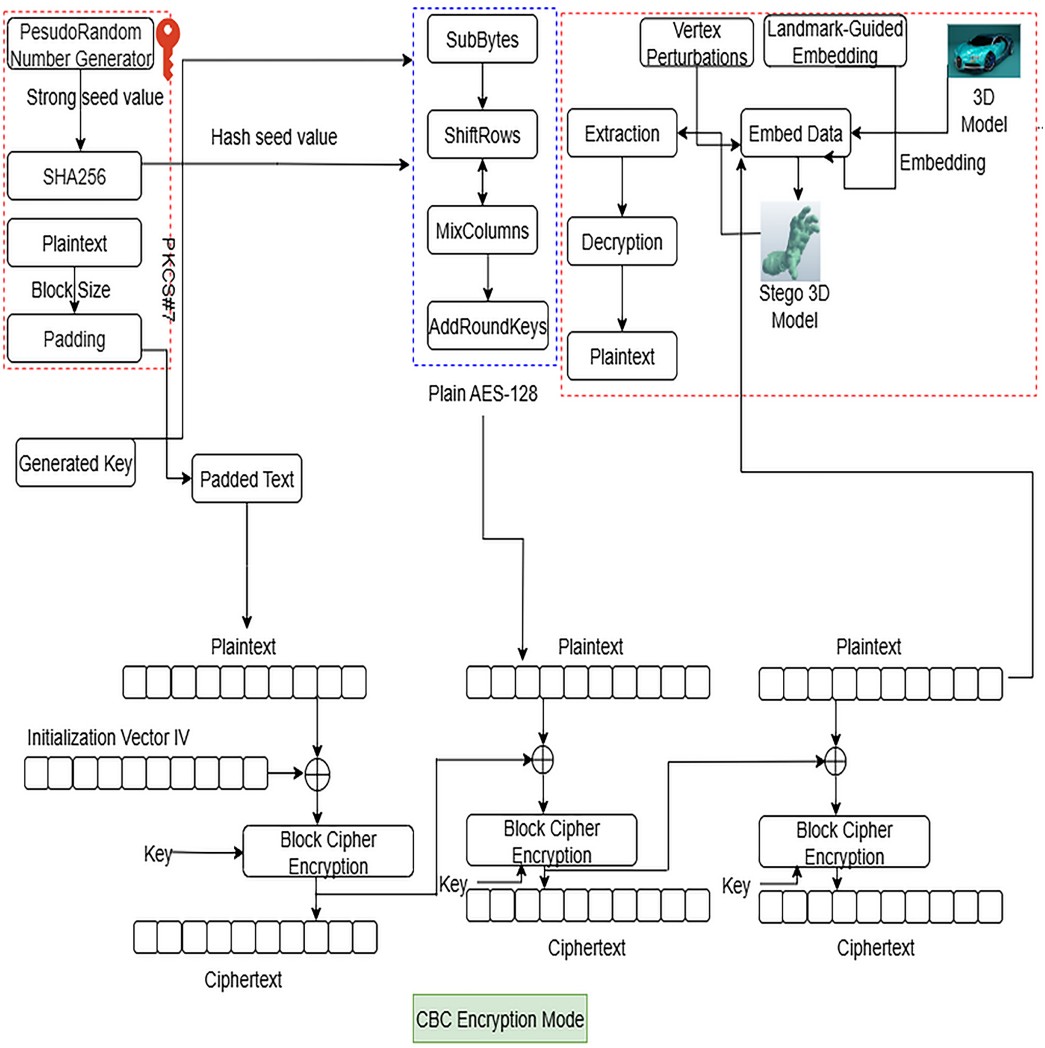

**Figure 1 Multi-layered framework with SHA256-based strong key, AES-128 using CBC mode reducing geometric inconsistency vulnerabilities.**

## Types of attacks on 3D model steganography

- Geometric feature-based attacks are forms of attacks that concentrate on uncovering nuanced variations in the morphology of the 3D model resulting from steganographic insertion.
- Texture-based attacks pertain to attacks that concentrate on modifying textures or other graphical characteristics of the 3D object.
- The statistical-based attacks are a type of attack that identifies deviations from the expected statistical properties of the 3D model.

## PROPOSED METHOD

This study integrated encryption and 3D steganography techniques to safeguard confidential data transmitted over an unsecured communication channel. Figure 1 depicts

an intricate multi-layered security structure that integrates cryptographic encryption with 3D steganography to safeguard sensitive data transfer. The system initiates with a comprehensive key creation procedure wherein a Pseudorandom Number Generator produces a resilient seed value, which is subsequently processed through SHA-256 hashing, integrating plaintext, block size, and padding parameters to yield a cryptographically safe key with augmented unpredictability. The framework utilizes AES-128 encryption in Cipher Block Chaining (CBC) mode, illustrated in the blue dashed section, where plaintext is segmented into blocks and subjected to multiple rounds of block cipher encryption. Each block employs an Initialization Vector (IV) and the generated key to produce ciphertext blocks that are contingent upon all preceding plaintext blocks, thereby augmenting security against pattern attacks. The encrypted ciphertext is then integrated into a 3D model utilizing advanced geometric steganography techniques illustrated on the right side of the figure. The system locates landmark regions inside the 3D facial model and executes vertex perturbations around these landmarks, employing landmark-guided embedding algorithms to embed the encrypted data into designated vertices while preserving the geometric integrity of the original model. This approach generates a "Stego 3D Model" that appears visually ordinary yet conceals encrypted information within its geometric framework. The system facilitates bidirectional operations, enabling both embedding and extraction processes, wherein authorized users can reverse the technique *via* decryption to retrieve the original plaintext from the stego 3D model.

## Encryption and decryption process

This study focused on the implementation of the AES algorithm in Cipher Block Chaining mode. We begin by initializing a new AES cipher in CBC mode. We subsequently produce a random initialization vector (IV) of 16 bytes. An initialization vector is employed to augment security and prevent the encryption of similar plaintexts into the same ciphertext. The encryption process generates encrypted data together with the IV (*Băneasă et al., 2024*). Decryption requires the IV in conjunction with the key to decipher the ciphertext. The key is crucial, as the security of AES encryption fundamentally depends on its confidentiality (*Boisrond, Tardif & Jaafar, 2024*). This research produced a robust and erratic key that enhanced security. The decryption function uses AES-128 in CBC mode to decipher the ciphertext. Similar steps are executed, including the creation of a new AES cipher object in CBC mode using the same secret key utilized for encryption. The algorithm subsequently decrypts the ciphertext utilizing the AES cipher object and the supplied IV. Ultimately, it eliminates the padding from the decoded data to retrieve the true plaintext. AES utilizes four repeated transformations for encryption and decryption.

- SubBytes
- ShiftRows
- MixColumns
- AddRoundKey transformations

Table 2 Key-block-round parameters by key size.

| No of bits | Length of key [$N_K$] | Size of block [$N_B$] | No of rounds[$N_R$] |
|---|---|---|---|
| 128 | 4 | 4 | 10 |
| 192 | 6 | 4 | 12 |
| 256 | 8 | 4 | 14 |

In the first round $r = 0$, only the addition of the round key is executed; in the last round $r = N_r$, the inv-/MixColumns operation is omitted. The keyschedule module expands the cipher key to $[N_r + 1] \times 4$ words of round keys. Each round utilizes a distinct 128-bit round key in the add round key function. The mix column is a crucial element that facilitates dispersion in AES. This enhances the security of AES against differential and linear attacks.

## CBC mode

The AES configurations are contingent upon the key size, as demonstrated in Table 2, which features a word size of 32 bits. Cipher Block Chaining (CBC) mode (*EITCA Academy, 2025*) is a crucial operational mode for block ciphers that improves the safety of encrypted data through the use of an IV. The initialization vector is essential for maintaining the security and integrity of the encryption process. In CBC mode, plaintext is divided into fixed-size blocks, each of which is encoded sequentially. The encryption of each block depends on the content of the current block and the ciphertext of the previous block. This dependence creates a sequential process, ensuring that identical unencrypted segments produce unique encrypted data blocks based on the use of a different IV for each encryption operation.

## Subtypes

The SubBytes transformation is an irregular byte substitution that operates independently on each bit of the current state using a substitution table. The S-Box is a $16 \times 16$ matrix comprising all 256 possible 8-bit values. It is utilized to modify the state nonlinearly, byte by byte. This invertible S-box is produced by combining two transformations (*William, 2008*):

- In the finite field GF[$2^8$], the multiplicative inverse of the element [00000000] is the element itself.
- Execute the subsequent affine transformation $(GF)^2$.

Algorithm 1 delineates the suggested enhanced AES algorithm. Integrating the mix column in the final cycle will augment the algorithm's security level. The inclusion of the MixColumn operation in the final round of the algorithm signifies improvements over conventional AES.

## Shift rows

The shift repeatedly shifts every single state row to the left side during cryptography. The uppermost row in the state is designated as $row[0]$, and the lowermost row is designated as

---

**Algorithm 1  AES algorithm.**

**Require:** Nr, Nb
**Ensure:** Cipher Text
    $data_X = \{\}$
    **for** int rd = 1; rd < Nr; rd++ **do**
        $s_1$ = SubBytes($data_X$, Nb)
        $s_2$ = ShiftRows($s_1$, Nb)
        $s_3$ = mixColumns($s_2$, Nb)
        $s_4$ = addRoundKey($s_3$, w, rd, Nb)
    **end for**
    $s_5$ = subBytes($s_4$, Nb)
    $s_6$ = shiftRows($s_5$, Nb)
    $s_7$ = mixColumns($s_6$, Nb)         ▷ Added mixColumns in the last round
    $s_8$ = addRoundKey($s_7$, w, Nr, Nb)
    **End**

---

$row[3]$. The ShiftRows operation executes an i-byte cyclical leftward shift on $row[i]$, where $i = 0, 1, 2, 3$.

## Mix columns

Each column of the present scenario can be interpreted as a four-term quadratic over $(GF(2^8))$. Each column is altered by integrating it with a predetermined equation.

## Add-round-key

The AddRoundKey operation integrates the Round Key into the state using a straightforward bitwise XOR process. Each round key comprises $N_b$ words obtained from the key scheduling process. The $N_b$ words are present in each column of the state, with $W_i$ denoting the key scheduling words, and "round" is a variable ranging from 0 to $N_r$. The initial cycle of key addition in the cipher occurs when rounding equals zero, before the first iteration of the rounding process. The Add Round Key transformation is executed during the $N_r$ cycles of the cipher, where $1 < round < N_r$ and $1 = round \times N_b$.

## Key generation

The study aimed to generate a cryptographically secure key by combining secure random number generation with SHA-256 hashing to improve the key's randomness and robustness. We generated a random integer comprising 256 bits, which our research transformed into a sequence of 32 bytes in big-endian byte order. This sequence served as the foundational element for the key generation process (*Alenizi et al., 2024*). We then generated a SHA-256 hash object, supplied the seed, and computed the seed's SHA-256 hash, yielding a 32-byte digest. Hashing introduces an element of unpredictability, enhancing the generated key's robustness against diverse attacks. The employment of a random seed, SHA-256 hashing, and an acceptable length for key generation produced a strong and unpredictable key suitable for security applications. Algorithm 2 illustrates the key scheduling technique that produces round keys for AES-128. The resultant key schedule comprises a linear array of four-byte words, represented as $[w_i]$, where i ranges from 0 to $N_b \times [N_r + 1]$. AES (*Daemen & Rijmen, 1999*) functions on 16-byte blocks.

---

**Algorithm 2  Expanding key.**

**Require:** byte key[$4 \times N_k$], word w[$N_b \times (Nr + 1)$], $N_k$
**Ensure:** Expanded key $w_d$

```
    word ← temp
    j = 0
    while j < N_k do
        w_d[j] = word(key[4*j], key[4*j+1], key[4*j+2], key[4*j+3])
        j = j + 1
    end while
    j = N_k
    while j < N_b × (N_r + 1) do
        temp = w_d[j-1]
        if j mod N_k = 0 then
            temp = SubWord(RotWord(temp)) ⊕ Rcon[j/N_k]
        else if N_k > 6 and j mod N_k = 4 then
            temp = SubWord(temp)
        end if
        w_d[j] = w_d[j-N_k] ⊕ temp
        j = j + 1
    end while
```

Consequently, padding is essential to guarantee that the data intended for encryption conforms to the requisite block size specifications. Appropriate padding mitigates some attack vectors and safeguards the integrity of the encrypted information.

## Secure key generation and data encryption in 3D steganography

AES-128 with CBC-IV and SHA-256 with salt and a 32-bit password fulfill different yet complementary functions in safeguarding data in 3D steganography. SHA-256 with salt and a 32-bit password is utilized for the secure production of the encryption key. This procedure incorporates the user's password, appends a random "salt" to it, and subsequently employs the SHA-256 cryptographic hash function. The resultant fixed-size hash output subsequently serves as the highly secure, 128-bit encryption key. The produced key is then utilized by the AES-128 algorithm functioning in Cipher Block Chaining (CBC) mode with an Initialization Vector (IV). AES-128 executes the encryption of sensitive plaintext data, converting it into ciphertext. The CBC mode guarantees that identical plaintext blocks yield distinct ciphertext blocks through its chaining method, while the IV introduces an initial layer of randomization, thwarting an attacker from inferring information from successive encryptions of identical material. SHA-256 fortifies the key, rendering it resilient against diverse attacks, while AES-128 with CBC-IV employs this secure key to encrypt the data, safeguarding its confidentiality before its concealment within the 3D model.

## EMBEDDING AND EXTRACTING PROCESS

We first removed extraneous vertices and faces. We subsequently performed the surface smoothing technique, resulting in the normalization of the 3D model for this study. Subsequently, we extracted geometric features and calculated the curvature, angles, and distances between different vertices. We choose the vertices based on their importance to

integrate the data into a 3D model. This study employed a smoothing ranking algorithm to prioritize the model's smooth and significant components above the rougher ones, as mesh inconsistency is more likely to occur in those areas. This study used Gaussian distortion to assess and analyze the model's texture, as it effectively represents the degree of homogeneity in localized regions. In a 3D segment of a singular ring situated at a specific vertex, the lengths of the adjacent sides of the central facet $Vt_i$ are $et(i, j)$ and $et(i, j + 1)$, where $et(j, j + 1)$ denotes the distance between $Vt_j^i$ and $Vt_{j+1}^i$. The uniformity of each vertex is determined by calculating the circumference of the triangular adjacent to facet $Vt_i$, referred to as $St(i, j, j + 1)$, and subsequently designating this area as $St_i$, as specified in Eq. (1).

$$St_i = \frac{\sqrt{qt_{j+1,j+1}^{(i)}(qt_{j+1,j+1}^{(i)} - et(i,j))(qt_{j+1,j+1}^{(i)} - et(i,j+1))(qt_{j+1,j+1}^{(i)} - et(j,j+1))}}{3} \tag{1}$$

where $qt_{j+1,j+1}^i$ represents the half circumference of the triangle patch, and the discrete Gaussian curvature of the triangular mesh, as determined from the integral Gauss-Bonnet formula, is articulated in Eq. (2):

$$Kt_i = \frac{2\pi - \sum_{rt=1}^{Rt} \theta_{rt}^{(i)}}{At_i}, \quad rt = 1, 2, \ldots, Rt. \tag{2}$$

$$\theta_{rt}^{(i)} = \arccos\left[\frac{(et(i,j))^2 + (et(i,j+1))^2 - (et(j,j+1))^2}{2et(i,j)et(i,j+1)}\right]. \tag{3}$$

The area of each triangular element in the grid is divided into three segments based on its vertices. To identify the region nearest to the center linked with the specific vertex $Vt_i$ in the mesh, the size of the one-ring neighborhood surrounding that vertex is computed. The area includes the Voronoi region and the centroid zone of each vertex within the radius, generally defined as the average edge length of the mesh. This region is delineated by Eq. (4).

$$At_i = \frac{1}{3}\sum_{rt=1}^{Rt} St_i. \tag{4}$$

## SHA256

The hash renders it computationally impractical to ascertain the original message. It accepts inputs of a specified size, producing a 256-bit hash. A minor modification to the provided data will produce a unique hash output. The SHA-256 algorithm (*Vaughn & Borowczak, 2024*; *Tran, Pham & Nakashima, 2021*) sequentially computes intermediate hash values for data segments. The hash output of the preceding data block serves as the principal reference for computing the hash of the subsequent block. The hash value of the complete text is regarded as the result of the final information block. It consists of two phases: data expansion (ME) and data reduction (MC). The process expands the 512-bit original text into 64 segments, each containing 32-bit data $W_i(0 \leq i \leq 63)$. In the first 16 rounds, the ME partitions the 512-bit input data into 16 segments of thirty-two bits each, denoted as $W_i$, where j varies from 0 to 15. In the final 48 rounds, the ME calculates

48 segments of 32-bit processing input $Wi$ for $16 \leq i \leq 63$ using Eq. (5). To calculate $W_i$ for $16 \leq i \leq 63$, three 32-bit adders and two logical operations, $0(z)$ and $1(z)$, are necessary.

$$Wr_i = \sigma_1(Wr_{i-2}) + Wr_{i-7} + \sigma_0(Wr_{i-15}) + Wr_{i-16}. \tag{5}$$

$0(z)$ and $1(z)$ are calculated as per Eqs. (6) and (7).

$$\sigma_0(z) = St^7(z) \oplus St^{18}(z) \oplus Rt^3(z). \tag{6}$$

Let $St_n(z)$ and $Rt_n(z)$ represent the right rotation and right shift of data $z$ by $n$ bits, respectively.

$$\sigma_1(z) = St^{17}(z) \oplus St^{19}(z) \oplus Rt^{10}(z). \tag{7}$$

The MC process generates a 256-bit hash value from the outputs of the ME process, consisting of 64 chunks of $Wr_j$, $0 \leq j \leq 63$).

$$Ti_1 = h_k + \Sigma_1(et) + Cj(et, fi, gl) + Ka_j + Wr_j. \tag{8}$$

The procedure comprises two primary phases: iterations and hash modifications.

$$Ti_2 = \Sigma_0(au) + Mag(au, bq, cd). \tag{9}$$

During the loop step, eight loop hash values (au, bq, cd, dr, et, fi, gl, and hl) are initialized using the initial hash values $H_0, H_1, \ldots, H_7$.

$$a = T_1 + T_2. \tag{10}$$

The loop hash values ab cdef gh are then computed and updated over 64 iterations.

$$e = d + T_1. \tag{11}$$

In each iteration loop j, $0 \leq i \leq 63$), by Eqs. (8)–(12) are implemented.

$$bq = au; cd = bq; dr = cd; fi = et; gl = fi; hl = gl. \tag{12}$$

Logical functions such as $0(x)$, $1(x)$, $C_h(x, y, z)$, and $M_{aj}(x, y, z)$ are computed using the Eqs. (13)–(16).

$$\Sigma_0(z) = St^2(z) \oplus St^{13}(z) \oplus St^{22}(z)ma_0(z) = St^2(z) \oplus St^{13}(z) \oplus St^{22}(z). \tag{13}$$
$$\Sigma_1(z) = St^6(z) \oplus St^{11}(z) \oplus St^{25}(z). \tag{14}$$
$$Ch(xi, yj, zk) = (xi \wedge yj) \oplus (\neg xi \wedge zk). \tag{15}$$
$$Ma(xi, yj, zk) = (xi \wedge yj) \oplus (xi \wedge zk) \oplus (yj \wedge zk). \tag{16}$$

During the hash update phase, the ultimate 256-bit hash value, segmented into eight segments of 32-bit data $H_O0, H_O1, \ldots, H_O7$, is derived by summing the initial hashes $H_0, H1, \ldots, H7$ as demonstrated in Eq. (17).

$$HO_0 = H_0 + a; \ldots; HO_7 = H_7 + h. \tag{17}$$

## SHA-256 hashed random seed for uniqueness and security analysis

The cryptographically secure hash method SHA-256 is utilized on this random seed to produce a fixed-length, unforeseeable, and distinct digest. This digest functions as a distinct identity for each embedded example and is subsequently employed to generate the

encryption key for AES-128 and to initiate the pseudo-random number generator that dictates the selection of embedding areas and patterns inside the 3D model. This guarantees that even for identical 3D models and concealed messages, each stego-model produced with a new random seed would display a distinct embedding arrangement, hence augmenting safety from statistical steganalysis and brute-force assaults on embedded locations. SHA-256-hashed random seeds provide strong security for 3D steganography by creating random patterns for choosing vertices that are hard to detect statistically. This study employed AES-256 in CBC mode to encrypt the confidential message, subsequently applying SHA-256 to hash a random seed that dictates which vertices will be altered to incorporate the encrypted data. The hash function guarantees that seed 1 generates a distinctly different embedding pattern from seed 2, rendering brute-force attacks computationally impractical. This cryptographic method spreads changes across the 3D mesh in a random way, making it harder for detection algorithms to find patterns. We obtained a limited number of the 3D models utilized in our tests from the publicly accessible repository at https://free3d.com/3d-models/. Before experimentation, all 3D models passed many preprocessing stages to guarantee consistency and appropriateness for our algorithms. This entailed (1) normalizing the models to conform to a unit bounding box centered at the origin, thus standardizing their scale and position. We transformed all polygon models into a solely triangular representation, as required by our vertex and face processing pipeline. We implemented a mesh cleaning algorithm to address common issues, such as non-manifold edges, redundant vertices, and degenerate faces, which helped maintain the integrity of the mesh structure.

## Preprocessing 3D model dataset

Preprocessing 3D model datasets is essential for guaranteeing data consistency, cleanliness, and optimization for machine learning applications. A methodology for concealing information in 3D models, grounded in grid saliency, introduces a preprocessing phase to streamline the 3D model and enrich it with a robust carrier through a multi-feature fusion technique. The preprocessing of three-dimensional models encompasses the following steps:

1. **Model simplification:** The model is streamlined by reducing its vertices and facets while preserving the fundamental geometric structure of the original model. The half-edge folding algorithm is employed for this purpose. The error cost is incorporated to mitigate the total mistake. This technique retains only the vertices that most effectively represent the model's primary information. The error cost associated with model simplification is expressed in Eq. (18):

$$\cot(u, v) = \max_{f \in T_u} \left\{ \min_{n \in T_{uv}} \sin^2 \frac{\alpha}{2} \right\} \times D \times \|u - v\| \tag{18}$$

   where $u$ and v are the two adjacent vertices of the original grid, v is the removed vertex. The degree of point v is denoted as D, and $\|u - v\|$ represents the length of the edge vu.

2. **Characterizing vertex smoothness:** The normal vectors of each grid vertex are calculated using the weighted mean of the points in its one-ring vicinity using Eq. (19).

$$\bar{n}\bar{v}_i = \frac{\sum_{j=1}^{N_i} v_j \in N_i \left( \overrightarrow{v_j} - \overrightarrow{v_m} \right)}{N_i} \tag{19}$$

where $N_i$ denotes the number of domain vertices $\bar{v}_j$ and $\bar{v}_m$ represents the average vector of all vertices at the center of the grid model. $\bar{n}\bar{v}_i$ denotes a normal vector to the vertex $v_i$.

3. **Defining the relative area of the grid:** In a triangular grid f within the model, its relative area is defined as the ratio of an individual area to the overall area. The overall surface area of the three-dimensional model is computed using Eq. (20).

$$S_i = \frac{1}{2} \left( \left| \overrightarrow{v_1 v_2} \right| \cdot \left| \overrightarrow{v_2 v_3} \right| \right). \tag{20}$$

The total surface area of the 3D model is in Eq. (21) as:

$$S = \sum_{\forall f \in M} S_i. \tag{21}$$

Given a sub-grid $f$, its proportional area is the proportion of one area to the overall area, computed using Eq. (22).

$$RS(f) = \frac{S_i}{S}. \tag{22}$$

4. **Defining boundary strength**. Perimeter strength is defined by a grid dihedral angle, which quantifies the bendable degree of an individual triangular grid to delineate concave and convex portions of the model. Based on the description of the dihedral angle, its computation is defined in Eq. (23).

$$\alpha = \pi + \text{sgn}\left( \cos\left( \left( \overrightarrow{c_1} + \overrightarrow{c_2} \right) \cdot \left( \overrightarrow{n_1} + \overrightarrow{n_2} \right) \right) \right) \times \arccos\left( \left( \overrightarrow{n_1} \cdot \overrightarrow{n_2} \right) / \left( \left| \overrightarrow{n_1} \right| \cdot \left| \overrightarrow{n_2} \right| \right) \right) \tag{23}$$

where $\overrightarrow{n_1}$ and $\overrightarrow{n_2}$ are the corresponding normal vectors of two planes created by the edges $e_{ij}$ of two nearby vertices $v_i$ and $v_j$ in the model. $c_1$ and $c_2$ are the midline vectors connecting the focal point of the shared edge $e_{ij}$ to another vertex of the adjacent surface.

5. **Enhancing parameters:** The three elements of vertex roughness, relative grid area, and boundary strength are integrated according to the local characteristics of the grid to establish local grid saliency. This integrated saliency metric is employed to partition the perceptual domain of the 3D model and identify high-energy embedding zones for information concealment. The local grid $PS(M_i)$ is defined as the linear weighted summation of vertex roughness, relative grid area, and boundary strength in Eq. (24).

$$PS(M_i) = \delta \cdot PD(v_i) + \gamma \cdot RS(f) + \eta \cdot \alpha \tag{24}$$

where $\gamma$, $\eta$, and $\alpha$ are the weight coefficients, with $\gamma$, $\eta$, and $\alpha$ $\varepsilon$ [0,1]. Additionally, constrain these three weight coefficients such that $\gamma + \eta + \alpha = 1$.

Algorithm 3 delineates the logical progression for ascertaining the saliency of vertices within a 3D model. The procedure is delineated into primary steps: determining vertex roughness, relative grid area, and boundary strength; subsequently amalgamating these attributes into a local saliency score; and lastly employing Mean Shift clustering to classify the vertices into Key, Secondary, and Normal Vertices.

**Algorithm 3** Calculate 3D model saliency.

**function** CALCULATE3DMODELSALIENCY (Model $V$)

**Require:** Input: Model $V$ (set of vertices $v_i = (x_i, y_i, z_i)$, $i = 1 \ldots N$)

**Ensure:** Output: Categorized Vertices (KV, SV, NV)

       $vertex\_roughness\_values \leftarrow empty\_list$

       $face\_relative\_areas \leftarrow empty\_list$

       $edge\_boundary\_strengths \leftarrow empty\_list$

       $local\_saliency\_scores \leftarrow empty\_list$

       $categorized\_vertices \leftarrow empty\_dictionary$

**Step 1: Define Vertex Roughness**

    **for** each vertex $v_i$ in $V$ **do**

        Determine normal vector $nv_i$ for $v_i$

        Details for one-ring neighborhood and weighted average

        Calculate $normal\_vector\_vi$ for $v_i$

        Add $normal\_vector\_vi$ to $vertex\_roughness\_values$

    **end for**

**Step 2: Define Relative Area of Grid**

    $total\_surface\_area \leftarrow 0$

    **for** each triangular face $f$ in Model **do**

        Let vertices of $f$ be $v_1, v_2, v_3$

        Calculate $area\_f = 0.5 \times |(v_2 - v_1) \times (v_3 - v_1)|$

        Cross product magnitude

        Add $area\_f$ to $face\_areas\_temp$

        $total\_surface\_area \leftarrow total\_surface\_area + area\_f$

    **end for**

    **for** each $area\_f$ in $face\_areas\_temp$ **do**

        $relative\_area\_f = area\_f / total\_surface\_area$

        Add $relative\_area\_f$ to $face\_relative\_areas$

    **end for**

**Step 3: Define Boundary Strength**

**for** each edge $e_{ij}$ connecting $v_i$ and $v_j$ in Model **do**

    Determine normal vectors $n_1, n_2$ of adjacent planes (faces) sharing $e_{ij}$

    Determine midline vectors $c_1, c_2$

    Calculate $dihedral\_angle\_alpha = formula\_from\_paper(n_1, n_2, c_1, c_2)$

    Add $dihedral\_angle\_alpha$ to $edge\_boundary\_strengths$

**end for**

**Step 4: Combine Features & Improve Parameters**

 Define weight coefficients $(\delta, \gamma, \eta)$ where $\delta + \gamma + \eta = 1$

$\delta \leftarrow 0.30$

$\gamma \leftarrow 0.25$

$\eta \leftarrow 0.20$

**for** each vertex $v_i$ in $V$ **do**

    Aggregate relevant roughness, relative area, and boundary strength for $v_i$

    Add $saliency\_score\_vi$ to $local\_saliency\_scores$

**end for**

**Step 5: Mean Shift Clustering Analysis**

Apply Mean Shift clustering to the local_saliency_scores

Categorize vertices based on cluster properties

Rule 1: Energy levels of KV, SV, NV decrease in order.

**for** each vertex $v_i$ with its $saliency\_score\_vi$ **do**

    **if** $saliency\_score\_vi$ belongs to a high-energy cluster **then**

        Assign $v_i$ to Key Vertices (KV)

    **else if** $saliency\_score\_vi$ belongs to a medium-energy cluster **then**

        Assign $v_i$ to Secondary Vertices (SV)

    **else**

        Assign $v_i$ to Normal Vertices (NV)

| Algorithm 3 | (continued) |
|---|---|

```
        end if
        Add v_i and its category to categorized_vertices
    end for
    return categorized_vertices
end function
```

**Table 3 Geometric dimensions of the 3D cover models.**

| 3D model | Geometry | Length (X) | Length (Y) | Length (Z) |
|---|---|---|---|---|
| Bugati | 231,572 × 3 | 58.35 | 43.55 | 43.57 |
| Liberty statue | 260,386 × 3 | 43.00 | 36.80 | 24.90 |
| Hand | 910,314 × 3 | 55.43 | 60.16 | 52.85 |
| Zombie hand | 97,618 × 3 | 17.10 | 19.00 | 13.96 |
| Harpy | 92,964 × 3 | 43.10 | 33.39 | 27.52 |

## 3D models

A 3D model consists of vertex information and facial information. The vertex information includes polygon sizes specified by Eq. (25).

$$Vt = \{Vt_i | 0 \leq i \leq Vt - 1\} \tag{25}$$

where $Vt = Xt_i, Yt_i, Zt_i \varepsilon \mathcal{R}^3$ and $Vt$ represents the quantity of triangles. The face pattern is denoted by Eq. (26).

$$\mathscr{F}\sqcup = \{ft_j | 0 \leq j \leq Ft - 1\} \tag{26}$$

where $Ft = Xt_j, Yt_k, Zt_l$ j,k,l $\varepsilon[0, 1, 2, \ldots, Vt - 1]$ and $Ft$ represent the number of faces.

This research presents a technique for 3D models in the OBJ file format. An OBJ format is a triangular representation of 3D surface geometry that delineates the surface structure of 3D objects, omitting other characteristics commonly found in standard 3D models, such as color. OBJ formats are exclusively utilized to depict closed surfaces or volumes. A dataset of 3D models in OBJ format is presented in Table 3.

## IEEE754 standard

The IEEE 754 Standard delineates two formats for floating-point representation: single precision and double precision, consisting of 32 and 64 bits, respectively. Both formats divide the bit allocation into three components: the sign bit, the exponent, and the mantissa. A 32-bit assembly of floating-point integers adheres to the IEEE 754 standard, comprising a sign bit, an 8-bit exponent, and a 23-bit mantissa. A floating-point number $\mathscr{L}$ can be expressed as $\mathscr{L} = [-1]^{Sn} \times Mn \times 2^{Ex-127}$. Specifically, Sn denotes the sign of the floating-point number, Ex signifies the exponent range of a floating-point number, which extends from −126 to 127, and Mn is utilized to store the mantissa. The pseudo-code for converting $\zeta t$ to the decimal floating-point value $\eta t$ is outlined in Algorithm 4, whereby $Vt(p, q)$ denotes each element from p to q of the vector $Vt$.

---

**Algorithm 4** Converting $\zeta t$ to $\eta t$.

**Require:** $\zeta t$
**Ensure:** $\eta t$
   $st = \zeta t(1:1)$
   $et = \zeta t(2:9)$
   $mt = \zeta t(10:32)$
   $\gamma t = 0$
   **for** $k = 1:1:23$ **do**
      $tn = \left(\frac{1}{2}\right)^k \times mt(k:k)$
      $\gamma t = \gamma t + tn$
   **end for**
   $\eta t = (-1)^{st} \times (1 + \gamma t) \times 2^{et-127}$
   **return** $\eta t$

---

# 3D STEGANOGRAPHY AND 3D MDOELS

In computer vision, prominence measurement emulates human attention to particular regions of an image, yielding salient areas that correspond with subjective evaluations. Assess the significance of a region or localized area in 3D grid modeling and network analysis analogous to human visual perception. *Yoo et al. (2022)* established the notion of model grid importance, which has been employed in several geometric computations, including form pairing, symmetry, contouring, and classification. Confidential information in 3D models is mostly integrated inside vertex coordinates and specified component locations. Segmenting 3D models into significant sub-regions according to perceptual relevance facilitates the incorporation of concealed data inside these areas, enhancing geographic symmetry, functionality, and the subtlety of the concealment technique.

## Embedding rules

This section utilizes the disparity in vertex data post-projection to incorporate confidential information. The particular concealment criteria are outlined below:

- Rule 1: The simulation's critical regions are segmented into x elements through the fusion of characteristics across different scales. Segmentation and vertex assessment in these regions are conducted utilizing the Average Shifting theory. Each area is categorized as Significant Vertices (IV), Additional Vertices (OV), or Routine Vertices (RV). The energies of IV, OV, and RV decrease sequentially according to the Rules of the data concealment energy theory.

- Rule 2: The initial concealed image was obscured by *De Simone, Gutiérrez & Le Callet (2019)* to generate the binary hidden sequence $B_i$, including $b_i$ bits. Task 1 indicates that the concealed series $B_i$ can be partitioned into $x_t$ segments, designated as $Bi_K^1, Bi_K^2, \ldots, Bi_K^{xt}$, arranged in descending order. The dimensions of each section are represented by $b_{i1}, b_{i2}, \ldots, b_{ixt}$, so that $b_{i1} + b_{i2} + \ldots + b_{ixt} = b_i$.

- Rule 3: By Rule 1 and Rule 2, the $OV$ of $xt$ areas is depicted on a two-dimensional plane. Arrange the diverse singular values in ascending order as $OV_1, OV_1, OV_1, \ldots, OV_{xt}$. Determine the disparity in the y-coordinate of the subkey position and integrate it by

correlating it with the segmented confidential data $Bi_K^1, Bi_K^2, \ldots, Bi_K^{xt}$. Upon the acquisition of confidential information, data from each segment $b_{i1} \approx b_{ixt}$ is amalgamated into a comprehensive concealed series B according to length.

- Rule 4: The confidential data is altered and concealed through the data-hiding technique by modifying the positional discrepancies within the complex $OV$. The numbers 0 and 1 denote the resulting positional discrepancy. The value of the difference is expressed by Eq. (27).

$$d = \bar{z}_{i+1} - \bar{z}_i \tag{27}$$

where, $\bar{z}_{i+1}, \bar{z}_i$ adjacent ordinate values.

Figure 2 depicts a detailed bidirectional workflow for the embedding and extraction of encrypted data within 3D models, showcasing the entire steganographic process from early model creation to final data recovery. The embedding process commences with comprehensive 3D model preprocessing, during which the original model is smoothed to enhance its geometric attributes, subsequently followed by the extraction of triangular patches that are the essential geometric components of the mesh structure. The system subsequently executes feature fusion to amalgamate pertinent geometric attributes and implements 3D model labeling to discern and classify various locations within the model, with particular emphasis on landmark areas conducive to data embedding. Mean shift clustering is utilized to categorize analogous geometric features and establish ideal embedding locations while preserving the model's structural integrity. The core embedding phase entails navigating and organizing the processed 3D model to systematically pinpoint relevant vertices for data insertion, accompanied by a corresponding data process that aligns the encrypted ciphertext with suitable embedding positions inside the model's geometric framework. The embedding algorithm meticulously integrates encrypted data into chosen vertices adjacent to landmark regions, employing geometric features to maintain the imperceptibility of the concealed information while safeguarding the model's visual integrity. This process employs Arnold scrambling techniques to augment security by randomizing the data placement pattern, thereby complicating detection or extraction of hidden information by unauthorized parties, ultimately resulting in a stego 3D model that resembles the original while containing concealed encrypted data. The extraction process replicates the embedding workflow in reverse, commencing with the stego 3D model and implementing identical preprocessing steps, which encompass triangular patch extraction, feature fusion, 3D model labeling, mean shift clustering, and traversing and sorting to ascertain the locations of the embedded data. The system subsequently does data matching to identify concealed information within the model's geometric framework, followed by the reversal of Arnold scrambling to restore the original data configuration. The extraction phase methodically obtains the embedded ciphertext from the designated vertices and implements the decryption process to restore the original plaintext, ensuring that authorized users can effectively access the concealed information while preserving the security and integrity of the steganographic communication system.

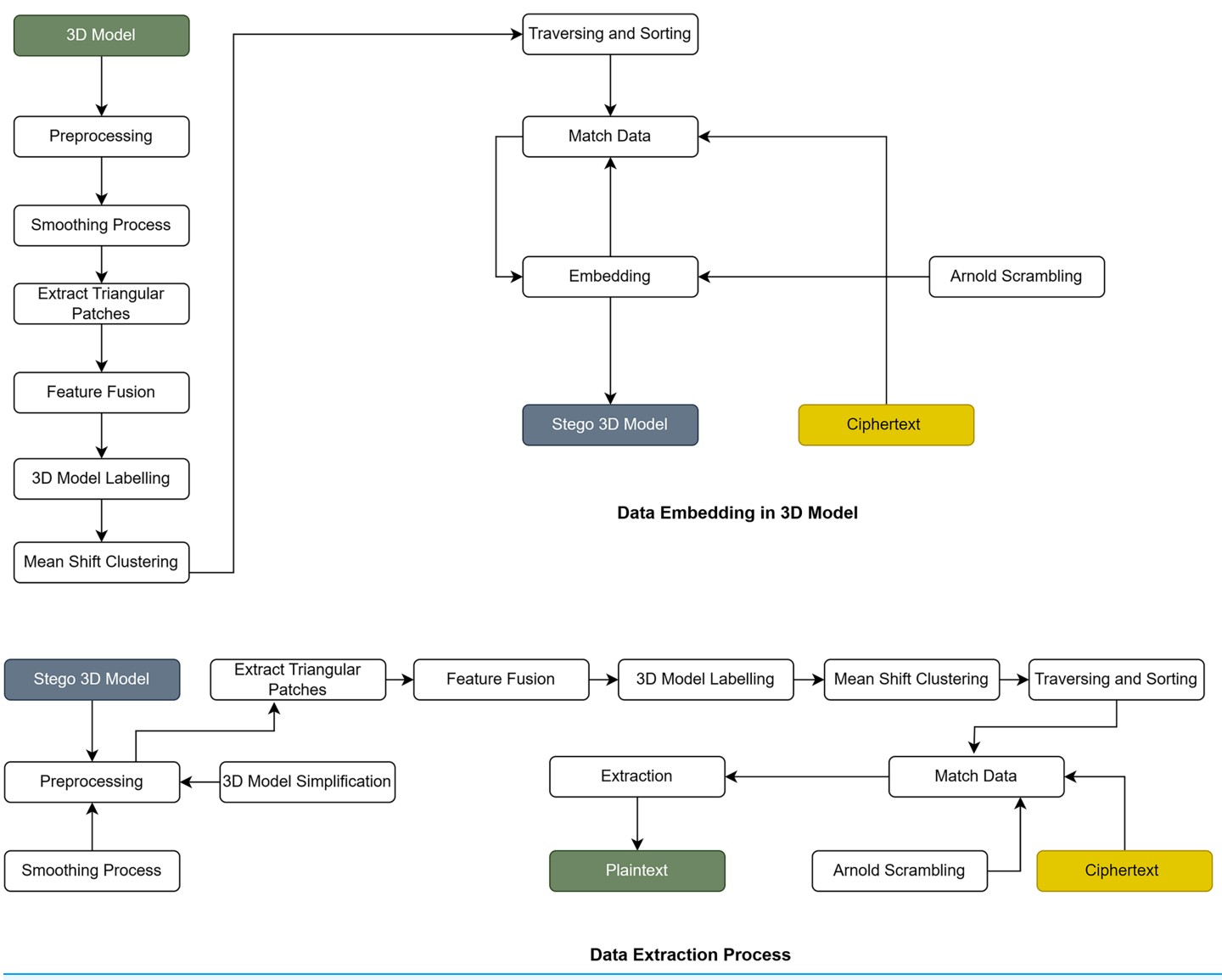

**Figure 2  Data embedding and extraction pipeline for 3D steganography.**

## Embedding data in 3D model

Effective 3D model annotation was achieved by calculating the local significance zone of the 3D model using the multiple scales local feature fusion method. Rule 1 was utilized to classify the region vertices into IV, OV, and RV. The preliminary phase involves streamlining the 3D model by curving its edges. To attain a 2D projection, OV coordinates within the area of interest are translated onto a two-dimensional plane. Define the triangular matrix of surfaces in the region as $St[St_1, St_2, St_3, \ldots, St_n]$, the vertex set as $Vt[Vt_1, Vt_2, Vt_3, \ldots, Vt_n]$, and characterize the standard vector of the area as the average standard vector for each triangular area. Designate the coordinates of a vertex as $Vt = [xt_i, yt_i, zt_i]$, and subsequently transform the 3D coordinate set $O-XtYtZt$ into a two-dimensional coordinate system $O-\bar{X}t\bar{Y}t\bar{Z}t$. Figure 2 depicts the flow of the procedure.

Identify the maximum values $\bar{y}t_{max}$ and the minimum values $\bar{y}t_{min}$ for the vertical coordinates $\bar{y}t$ within the two-dimensional coordinates $\bar{V}t$. Execute the Arnold transformation, a two-dimensional staggering technique frequently employed in image encryption, on the concealed image $Nt \times Nt$ to obtain the scrambled binary hidden sequence $Bt$. Ultimately, incorporate the confidential data by executing differential conversion to the binary data, as stipulated by Rule 3. Comparing the preliminary modified data with the confidential data series B. If they diverge, promptly rectify the discrepancy in values by adjusting the ordinate values $\bar{z}_{i+1}$ and $\bar{z}_i$, ensuring they remain within the permissible range of 0 to 1. Adjust the designated area to incorporate $bt$ bits of confidential information by utilizing the previously mentioned methods. Reconstruct the 3D model to acquire the secret-embedded model and finalize the data concealment; the two-dimensional coordinate refraction, which harbors concealed information, is manifested in the 3D model.

## Extracting data from 3D models

The following rules are crucial for the extraction of sensitive data, representing an inversion of the embedding process. To acquire 3D matrix regions of significant value, compact matrices were segmented utilizing the multi-scale fusing technique. A widely used clustering technique known as the Average Transition hypothesis is employed to categorize the IV, OV, and RV components of the split compact vectors into separate groups. Represent the operating parameters of each area on a two-dimensional coordinate system. Arrange the two-dimensional points $\bar{x}t$ and $\bar{y}t$ based on their cumulative vertical and horizontal positions in descending order, and subsequently calculate the resulting values $d_{xt}$ and $d_{yt}$ for neighboring coordinates. Assign a value of 0 or 1 based on the threshold of the respective interval difference, adhering to the Rules of gap-expressing conversion. An equivalent quantity of bits can be embedded within the same space to ascertain the integrity of the confidential information. Extract and integrate sensitive data from many sites into the final confidential data collection. If $Bi_K = Bt$, it indicates that the model is impervious to an attack or that the attack lacks the requisite intensity to jeopardize the extraction of secret information, hence affirming the model's resilience against such an onslaught. If not, the model has been penetrated, leading to a defined extent of damage and requiring the fortification of the algorithm's resilience against this attack in subsequent iterations.

# PERFORMANCE EVALUATION METRICS

The reliability essentially assesses the algorithm's ability to acquire crucial details for uncovering hidden confidential information against widespread image attacks holistically.

## Bit correct rate

The bit correct rate (BCR) functions as the assessment metric as specified in Eq. (28). In this situation, $No_i$ represents the precise quantity of retrieved confidential data, but $No_j$ indicates the entire volume of concealed data.

$$B_{CR} = \frac{No_i}{No_j} \times 100\%. \tag{28}$$

### Region Hausdorff distance

The Region Hausdorff Distance (RHD) is a crucial metric for assessing the geometric distortions induced by steganographic embedding in specific regions of 3D models and is defined in Eq. (29):

$$R_{HD} = \max\{\max\{\min_{p \in R_1, q \in R_2} \text{Norm}[p-q]\}, \max\{\min_{p \in R_2, q \in R_1} \text{Norm}[p-q]\}\}. \tag{29}$$

### Correlation coefficient

The correlation coefficient (CC) (*Kaleem, Malik & Sajid, 2025*) is employed to assess the significance of the results. In Eq. (30), the cover 3D model is represented by $c$ and the stego 3D model by $d$. The average of the stego 3D model is designated as $\bar{d}$, whereas the mean of the cover 3D model is represented as $\bar{c}$.

$$CC[c,d] = \frac{\sum_{k=1}^{M_i}[d_j - \bar{d}][c_j - \bar{c}]}{\sqrt{\sum[d_j - \bar{d}]^2 \sum[c_j - \bar{c}]^2}} \tag{30}$$

### Peak signal to noise ratio

Peak signal to noise ratio (PSNR) (*Wu & Li, 2024*) is a recognized metric employed to evaluate the quality of steganographic embedding in 3D models. It quantifies the ratio of the signal's maximal potential power to the power of disruptive noise that affects its representation and is articulated by Eq. (31):

$$P_{snr} = 10 \log_{10} \frac{W_i \times H_j \times 255^2}{\sum_{c=1}^{W_i} \sum_{d=1}^{H_j}[Co_{cd} - St_{cd}]^2} \tag{31}$$

### Root mean squared error

Root mean squared error (RMSE) (*Kaleem, Malik & Sajid, 2025*) quantifies the variance per pixel resulting from processing. Pixel variations diminish when RMSE values drop. RMSE is calculated by Eq. (32).

$$RMSE = \sqrt{\frac{I}{mn} \sum_{i=0}^{m-1} \sum_{j=0}^{n-1} [C[Im, jk] - S[Im, jk]]^2} = \sqrt{MSE}. \tag{32}$$

### Mean squared error

An ideal steganographic method would yield a mean squared error of zero, indicating that no discernible modifications are made after embedding. The mean squared error (MSE) can be computed utilizing Eq. (33).

$$M_{SE} = \frac{1}{K} \sum_{i=1}^{K} (x_i - \hat{x}_i)^2. \tag{33}$$

| Table 4 Experimental setup. | |
|---|---|
| **Parameters** | **Operators** |
| GPU | T4 (Google Colab) |
| System RAM | 12.7 GB |
| GPU RAM | 15.0 GB |
| Disk | 112.6 GB |
| Language | Python |
| Video memory | 24 GB |

## Earth mover's distance

Earth mover's distance (EMD), or Wasserstein distance, quantifies the least expense required to convert one probability distribution into another. In the realm of 3D point clouds, it considers each point cloud as a distribution of Earth Mover's Distance that must be adjusted to align with another distribution. EMD is often governed by global distribution while accounting for the faithfulness of intricate structures, which corresponds effectively with human perception of 3D form similarity. EMD evaluates the best correspondence across all points, rendering it less susceptible to isolated outliers in synthetic datasets. For two point clouds P and Q, the EMD addresses the best possible transfer challenge defined in Eq. (34).

$$EMD(P, Q) = \min \sum_{i,j} \|p_i - q_j\| \cdot f_{ij}. \tag{34}$$

min denotes the minimization operator $\sum_{i,j}$ represents the double summation over indices i and j $\|p_i - q_j\|$ signifies the norm (distance) between points $\cdot$ indicates the multiplication dot $f_{ij}$ refers to the flow variable with subscripts.

## RESULTS AND DISCUSSION

Comprehensive reference metrics utilize a reference 3D model to provide a direct quality comparison. Without a reference, reference metrics utilize expected image statistics to calculate quality scores.

### Experimental settings

The experiments were conducted using the specified parameters, with the details of the experimental configuration outlined in Table 4. The current study discovered the optimal hyperparameter configurations through a comprehensive exploration of numerous combinations, as presented in Table 5.

### Dataset

This study introduces an integrated method for 3D models in the OBJ file format. Figure 3 depicts a compilation of 3D models in the OBJ file format. Table 6 outlines the model specifications, including the number of vertices and faces.

**Table 5 Hyperparameter configuration of the proposed model.**

| | |
|---|---|
| 3D model | 9 |
| Stego 3D model | 9 |
| Cryptography | Optimized CBC based AES-128 |
| Hash algorithm | SHA-256 |
| Embedding rules | ROI & saliency features |
| Key length | 32 Bytes |

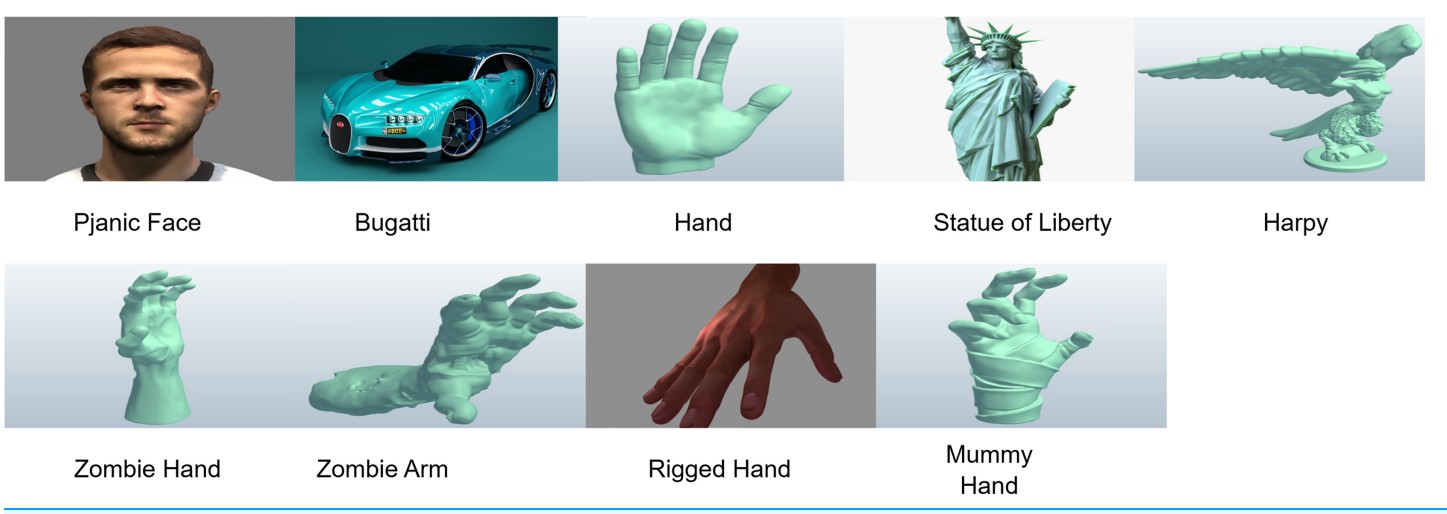

Pjanic Face    Bugatti    Hand    Statue of Liberty    Harpy

Zombie Hand    Zombie Arm    Rigged Hand    Mummy Hand

**Figure 3 3D models used for experimental studies.**

**Table 6 Geometric properties of the 3D model dataset.**

| Dataset | No. of vertices | No. of faces | Surface area | Volume |
|---|---|---|---|---|
| Bugati | 12,292 | 23,680 | 8.30 | 61.61 |
| Liberty statue | 514 | 1,024 | 0.01 | 0.00 |
| Hand | 66,848 | 133,692 | 511.03 | 442.98 |
| Zombie hand | 17,961 | 35,772 | 748.71 | 485.02 |
| Harpy | 66,143 | 131,510 | 335.20 | 57.50 |
| Pjanic face | 220,141 | 431,544 | 58.80 | 28.32 |
| Rigged hand | 1,071 | 2,072 | 0.10 | −0.00 |
| Mummy hand | 49,098 | 97,984 | 542.97 | 306.96 |
| Zombie arm | 37,472 | 74,770 | 673.32 | 666.93 |

## Experimental results

Table 7 displays experimental results for multiple datasets across several assessment metrics: CC, PSNR, MSE, RHD, BCR, and EMD. Typically, elevated CC, PSNR, and BCR values signify superior performance, whereas diminished MSE, RHD, and EMD values are

**Table 7 Experimental results based on different evaluation metrics.**

| Dataset | CC | PSNR | MSE | RHD | BCR | EMD |
|---------|------|-------|-------|------|-------|------|
| Bugati | 0.975 | 62.19 | 13.36 | 0.16 | 45.69 | 0.61 |
| Liberty statue | 0.993 | 54.39 | 11.65 | 0.35 | 40.85 | 0.56 |
| Hand | 0.948 | 52.34 | 12.87 | 0.31 | 42.25 | 0.62 |
| Zombie hand | 0.957 | 58.90 | 23.98 | 0.26 | 46.77 | 0.54 |
| Harpy | 0.999 | 66.17 | 6.22 | 0.03 | 56.38 | 0.67 |
| Pjanic face | 0.950 | 50.40 | 3.77 | 0.08 | 33.08 | 0.83 |
| Rigged hand | 0.813 | 34.54 | 5.94 | 0.19 | 25.31 | 0.90 |
| Mummy hand | 0.906 | 41.83 | 12.45 | 0.28 | 41.35 | 0.26 |
| Zombie arm | 0.941 | 46.21 | 7.25 | 0.10 | 33.83 | 0.89 |

favored. The Bugatti and Liberty Statue datasets exhibit robust performance, characterized by elevated CC and PSNR values, alongside comparatively low MSE and EMD metrics. The Harpy dataset has remarkably high CC (0.999) and PSNR (66.17), alongside the lowest MSE (6.22) and RHD (0.03), indicating outstanding performance. Conversely, Rigged Hand demonstrates the lowest CC (0.813) and PSNR (34.54), alongside the highest EMD (0.90), signifying inferior performance relative to the others. The Pjanic Face exhibits a comparatively low PSNR of 50.40 and a high EMD of 0.83. Mummy Hand exhibits a remarkably low EMD (0.26) despite average performance in other measures, but Zombie Arm and Zombie Hand display similar outcomes, both achieving satisfactory CC and PSNR values. The data indicates a differential performance among datasets, with certain datasets surpassing others according to the aggregated metrics. The suggested 3D steganography method calculates the RHD to assess the maximum distortion at a regional level inside the 3D model after data embedding. An optimal steganographic method should aim for the minimal possible RHD, indicating reduced localized distortion and enhanced fidelity between the cover and stego 3D models. The RHD study confirms that the regional alterations in our steganographically embedded 3D models remain beyond detectable thresholds, thus maintaining both the model's integrity and the imperceptibility of the embedded data.

Figure 4 illustrates the efficacy of various 3D models across multiple evaluation metrics: CC, PSNR, MSE, RHD, BCR, and EMD. The experimental results indicate that datasets such as "Harpy" and "Bugati" exhibit robust performance, evidenced by elevated PSNR values (red bars) and comparatively low MSE (green bars) and RHD (yellow bars), implying superior reconstruction quality and minimal mistakes. In contrast, "Rigged Hand" regularly demonstrates inferior PSNR and elevated EMD (orange bars), signifying subpar performance relative to other models. "Zombie Hand" and "Mummy Hand" exhibit moderate PSNR, although their EMD values vary, with "Mummy Hand" displaying a significantly lower EMD than both "Zombie Hand" and "Zombie Arm." The "Pjanic Face" dataset has a reduced PSNR and elevated EMD, akin to "Rigged Hand," indicating difficulties in its assessment. The graph adeptly illustrates the diverse strengths and shortcomings of each 3D model across many evaluation parameters, identifying "Harpy"

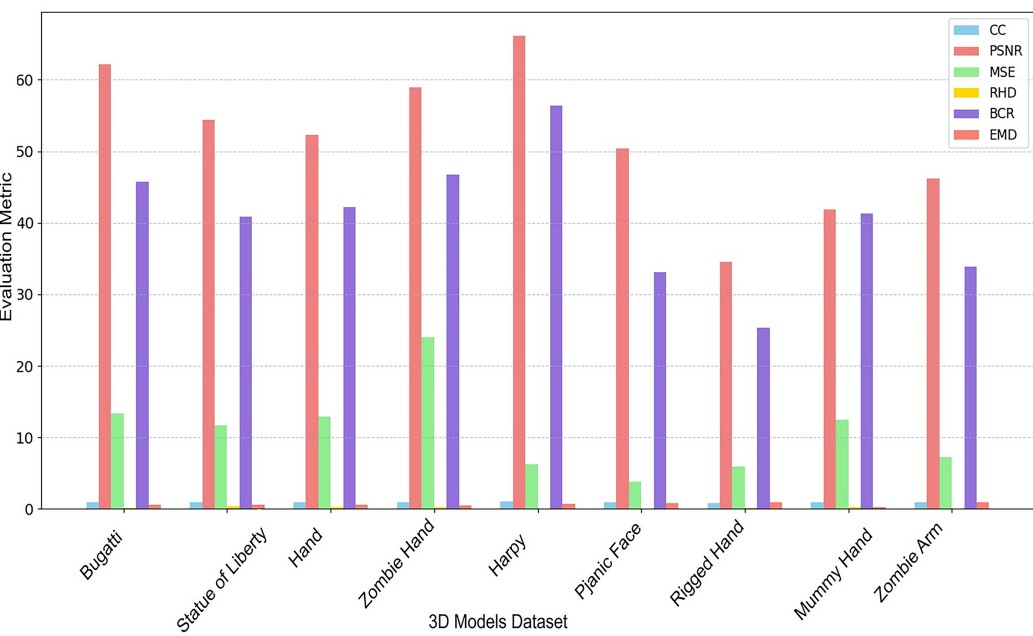

**Figure 4** Robustness and capacity evaluation of 3D steganography techniques.

as a leading performer and "Rigged Hand" as a model with considerable potential for enhancement. The experiments indicate that the suggested model achieved enhanced correlation across all datasets. The effectiveness of the proposed approach varies significantly based on supplementary metrics, including PSNR, RHD, MSE, BCR, and EMD. The Harpy dataset performed better across most criteria, while the Zombie Hand frequently displayed marginally inferior performance.

Table 8 presents the insertion and extraction durations for various text sizes achieved through experimentation. The insertion time often escalates with an increase in text size. Table 8 displays the encryption and decryption durations in seconds for a sequence of nine operations, referred to as "Number of iterations." A discernible trend indicates that both encryption and decryption durations typically escalate as the number of iterations advances, implying that the complexity or volume of the data being processed intensifies with each successive operation. "Iteration 1" exhibits the minimal encryption duration of 0.021 s and a decryption duration of 0.0023 s, whereas "iteration 8" demonstrates the maximal encryption duration of 0.7913 s. Throughout all procedures, the encryption duration is continuously and markedly greater than the decryption duration. This discrepancy is especially evident in subsequent operations, where encryption durations are approximately twice as long as decryption durations; for instance, in "iteration 7," encryption takes 0.7809 s compared to decryption at 0.491 s. This persistent disparity indicates that the encryption method is more computationally demanding or entails more stages than the decryption process for a specific 3D model.

Figure 5 depicts the performance of encryption and decryption operations over nine different iterations, indicating the duration of each operation in seconds. A discernible

**Table 8 Time complexity measurements for cryptographic operations (sec).**

| Sr.# | Encryption time (sec) | Decryption time (sec) |
|---|---|---|
| 1 | 0.021 | 0.0023 |
| 2 | 0.067 | 0.016 |
| 3 | 0.1501 | 0.0270 |
| 4 | 0.1602 | 0.0356 |
| 5 | 0.1859 | 0.0510 |
| 6 | 0.7694 | 0.323 |
| 7 | 0.7809 | 0.491 |
| 8 | 0.7913 | 0.389 |
| 9 | 0.7817 | 0.415 |

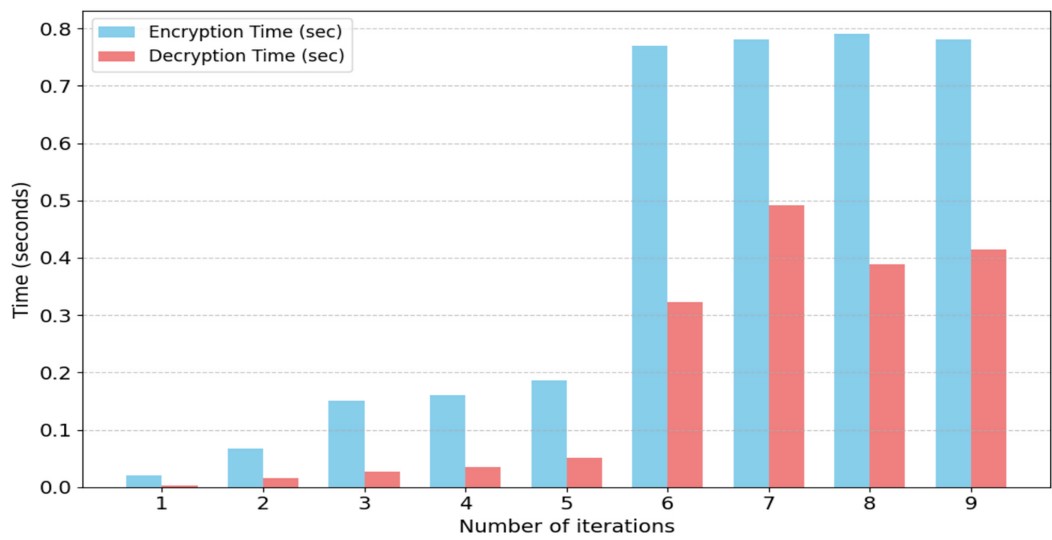

**Figure 5 Evaluation of computational efficiency in 3D steganography.**

trend is evident as the number of iterations escalates, with both encryption and decryption durations exhibiting an increase. Visualizing encryption and decryption provides critical insights into the efficiency and scalability of the system, identifying areas for optimization, especially in encryption, and underscoring the significance of resource management when managing larger datasets in response to the growing demand for extensive 3D model datasets.

Figure 6 depicts a comparison of encryption and decryption durations (in seconds) across nine repetitions. Both encryption and decryption durations exhibit an upward trend as the number of repetitions increases, signifying that the computational effort for both procedures typically escalates with additional operations. Nonetheless, a notable discrepancy is evident: encryption time (blue line) is continuously and significantly greater than decryption time (red line) throughout all repetitions. The disparity becomes more evident after the fifth iteration, as encryption time sharply escalates from around 0.18 s at iteration 5 to about 0.76 s at iteration 6, then stabilizes at an elevated level. The decryption

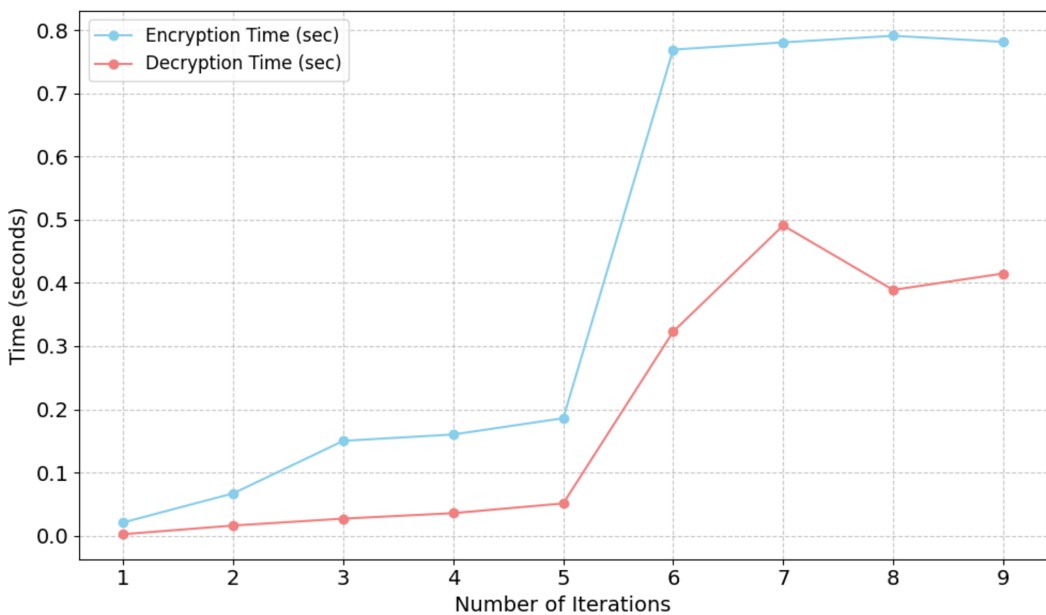

**Figure 6 Execution time analysis for data embedding and extraction.**

duration similarly escalates within this range, albeit at a far slower pace, reaching a maximum of approximately 0.49 s at iteration 7 before seeing a tiny decline. The consistently prolonged encryption duration indicates that the encryption algorithm is more computationally demanding or intricate than the decryption procedure, necessitating greater processing resources or steps every iteration.

## Entropy and resistance to side-channel attacks

Our experimental results show that the new steganographic method provides strong cryptographic security for all tested 3D models, with many important improvements compared to existing methods as shown in Table 9. The nearly perfect entropy values (7.963–7.988 bits/byte) for all models show that the method is better at resisting statistical analysis attacks, and the tiny timing differences (0.0087–0.0916 $\mu s$) provide strong protection against timing side-channel attacks, which is an important advance over traditional adaptive embedding techniques that often have weaknesses based on their structure. The very limited power usage (5.1–10.8 mW) and few cache misses (89–105) further validate the method's robustness against power analysis and cache-timing attacks. The Liberty Statue model demonstrates the approach's superior performance with exceptional metrics in all areas (entropy: 7.988, timing: 0.0593 $\mu s$, power: 9.7 mW), although the somewhat slower Zombie Hand variant (0.0916 $\mu s$) still achieves best-in-class entropy (7.989 bits/byte). The results show that our method successfully keeps security performance separate from the model's features, providing reliable protection no matter how complex the shape is, which is essential for using 3D steganography, where security should not depend on the content. The uniformity of these security assurances across many models signifies a considerable enhancement compared to earlier

**Table 9 Analysis of entropy and resistance to side-channel attacks.**

| Model | Entropy (bits/byte) | Timing variance (μs) | Power consumption (mW) | Cache misses |
|---|---|---|---|---|
| Pjanic face | 7.982 | 0.0447 | 11.3 | 98 |
| Bugati | 7.963 | 0.0124 | 8.1 | 105 |
| Liberty statue | 7.988 | 0.0593 | 9.7 | 89 |
| Rigged hand | 7.971 | 0.0581 | 9.9 | 94 |
| Hand | 7.967 | 0.0256 | 10.8 | 100 |
| Zombie hand | 7.975 | 0.0916 | 5.1 | 100 |

methods that frequently showed notable performance discrepancies influenced by input attributes.

## Histogram analysis of 3D models

We assess the 3D models of Bugati, Liberty Statue, Hand, Zombie Hand, and Harpy. Figure 7 depicts the distributions of face area and face angle. The algorithmic analysis retrieves significant information from the 3D models, illustrating the effectiveness of the proposed method. Harmonic assessment can provide a dispersion of empirical coordinate values in 3D model encryption approaches. The standard text graphs of the 3D models are illustrated in Fig. 7, whereas the encrypted graphs are depicted in Fig. 8. Figures 7 and 8 illustrate that the original text graphs display irregular distribution patterns, whereas the encrypted text graphs show uniform patterns; therefore, an intruder is unable to extract substantial information from the encrypted text.

## Performance evaluation in terms of geometric and visual quality

The present study used RMSE and RHD to quantitatively analyze the disparity between $M$ and $M_{level}$, where $M_{level} \in [Max, Avg, Min]$, in order to evaluate the geometric deformation of 3D models. We first compute the RMSE and RHD metrics by comparing the original 3D model Harpy $M$ with the 3D model $M'$, while varying the parameters p, q, and r. The data are illustrated as curves below to emphasize the increasing trend of geometrical deformation. Table 10 illustrates that only the *P*-value exhibits a linear increase, but the RMSE and RHD rates for $M$ and $M_{Max}$ remain relatively constant, despite a significant decline in RMSE and RHD rates for $M$ and both $M_{Avg}$ and $M_{Min}$ from their previously elevated values. Comparable incremental variations in $M_{Max}$ are also evident in Table 10 and Fig. 9. Concurrent inspection of the data at appropriate and intermediate phases reveals that the RMSE and RHD readings between $M$ and $M_{Avg}$ are nearly identical. This discovery relates to the facts concerning transparency and minimal levels for $p + q = 2 \times \alpha + 1$. The RMSE and RHD values between $M$ and $M_{Min}$ converge with those between $M$ and $M_{Trans}$ at $\alpha = \frac{p+q}{2}$, indicating that $Min(p, q, r) \approx [trans, \alpha]$.

## Comparative analysis of proposed 3D steganography and state-of-the-art techniques

Table 11 delineates the comparative outcomes based on the suggested methodology and encrypted approaches for 3D models. Unlike the approach adopted in *Gao et al. (2023)*,

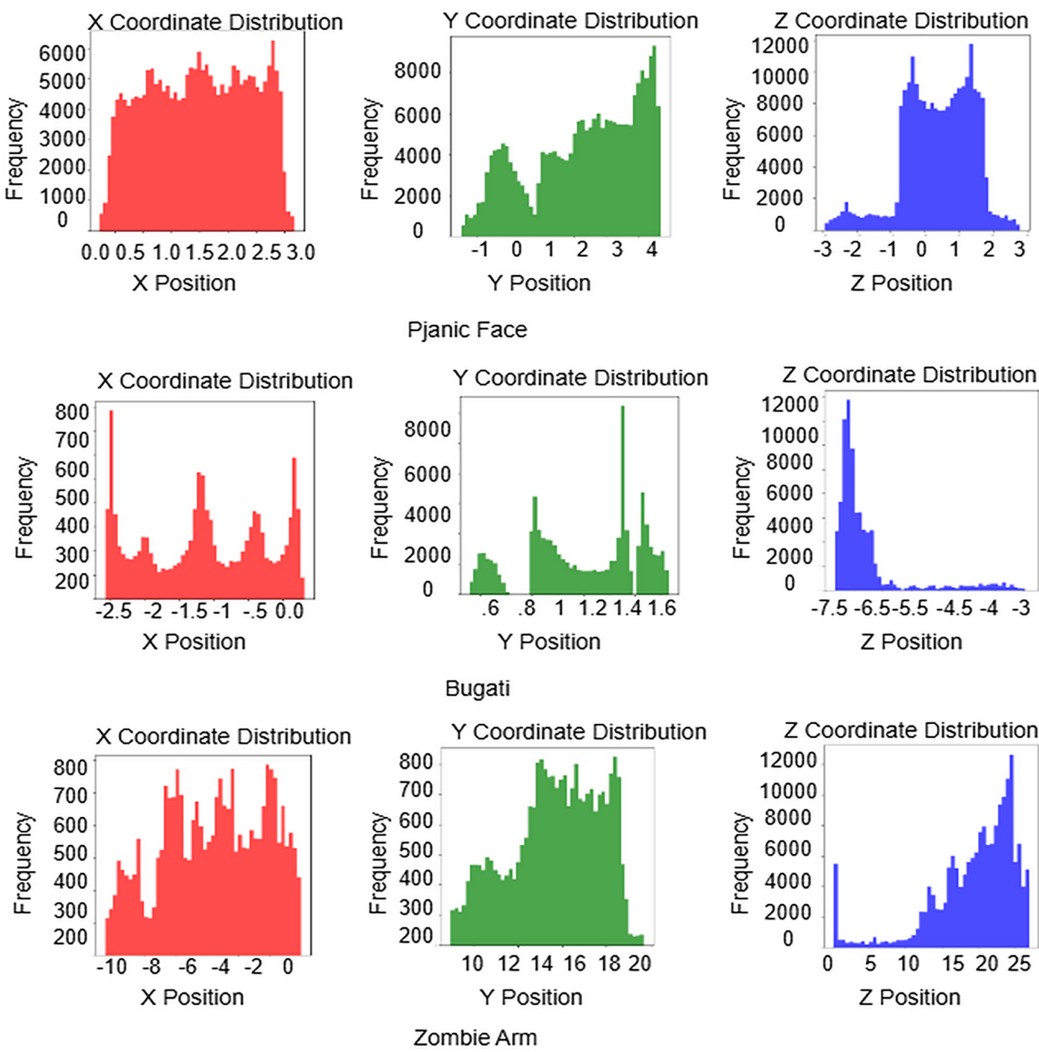

**Figure 7 Distribution of geometric properties across the 3D models.**

which utilizes a masked procedure to ascertain bit positions and spans before encrypting our methodology, like that in *Kostiuk et al. (2022)*, segments the mantissa bits of vertex coordinates into three sequential sub-blocks. The sub-blocks are then employed in block generation and then encrypted. Concerning the protected information, *Liang, He & Li (2019)* guarantees the authenticity of the connectivity of the 3D model's areas. Concurrently, *Wang, Xu & Li (2019)* and this study methodically retains the geometrical data associated with 3D models, encompassing aspects such as shape, content, and features of excellent quality. This technological study is the sole method that enables dynamic deciphering of the secured 3D model, as shown in Fig. 10.

## Unified average change intensity and number of pixels change rate

The Unified average change intensity (UACI) and number of pixels change rate (NPCR) are frequently utilized as metrics for assessing secret key sensitivity

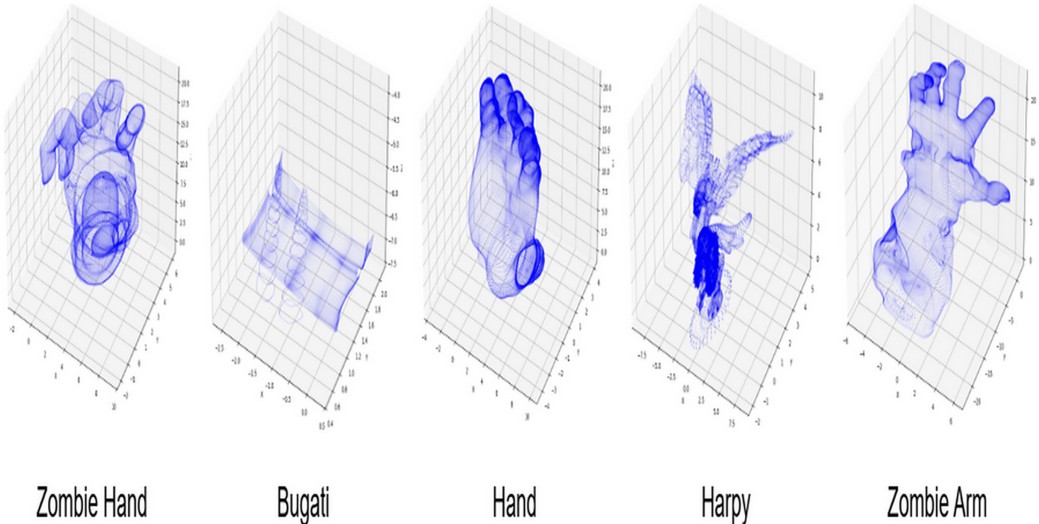

| Zombie Hand | Bugati | Hand | Harpy | Zombie Arm |

**Figure 8 Histogram uniformity analysis for encrypted 3D models.**

**Table 10 Quantitative evaluation of geometric fidelity with RMSE and RHD.**

| Visual level | RMSE/RHD | [1,6,9] | [2,6,6] | [3,6,9] | [4,6,9] | [5,6,9] |
|---|---|---|---|---|---|---|
| Max | RMSE | 45.95 | 45.87 | 45.88 | 45.94 | 45.90 |
| | RHD | 97.27 | 98.40 | 93.88 | 93.89 | 94.49 |
| Avg | RMSE | 21.27 | 13.83 | 15.46 | 12.85 | 3.24 |
| | RHD | 38.87 | 21.22 | 10.12 | 8.54 | 13.66 |
| Min | RMSE | 1.60 | 1.49 | 1.13 | 1.89 | 0.033 |
| | RHD | 0.84 | 1.92 | 1.49 | 1.24 | 1.16 |

(*Wu, Noonan & Agaian, 2011*). When UACI nears 33.33% and NPCR approaches 99.99%, the method demonstrates considerable robustness against differential attacks. The NPCR, described by Eq. (35), and the UACI, defined by Eq. (36), are computed as follows:

$$N_{PCR} = \frac{\sum_{k=1}^{m_i} \sum_{l=1}^{n_j} Ct(i,j)}{m_i \times n_j} \times 100\% \tag{35}$$

$$U_{ACI} = \frac{\sum_{k=1}^{m_i} \sum_{l=1}^{n_j} |\bar{C}t(i,j) - Ct(i,j)|}{m_i \times n_j \times 255} \times 100\% \tag{36}$$

where Eq. (37) or shows some conditional settings.

$$Dt(x,y) = \begin{cases} 1, & \text{if } \bar{C}t(x,y) \neq Ct(x,y) \\ 0, & \text{if } \bar{C}t(x,y) = Ct(x,y) \end{cases}. \tag{37}$$

Table 12 presents the experimental results comparing the NPCR and UACI values obtained from the research. All 3D models exhibited consistently high NPCR ratings. The values range from 37.34 to 31.32. It indicates a minimal likelihood that two adjacent vertices possess identical values post-encryption. The encrypted 3D model has significant volatility and dispersion. This study yielded UACI metric values ranging from 99.95 to

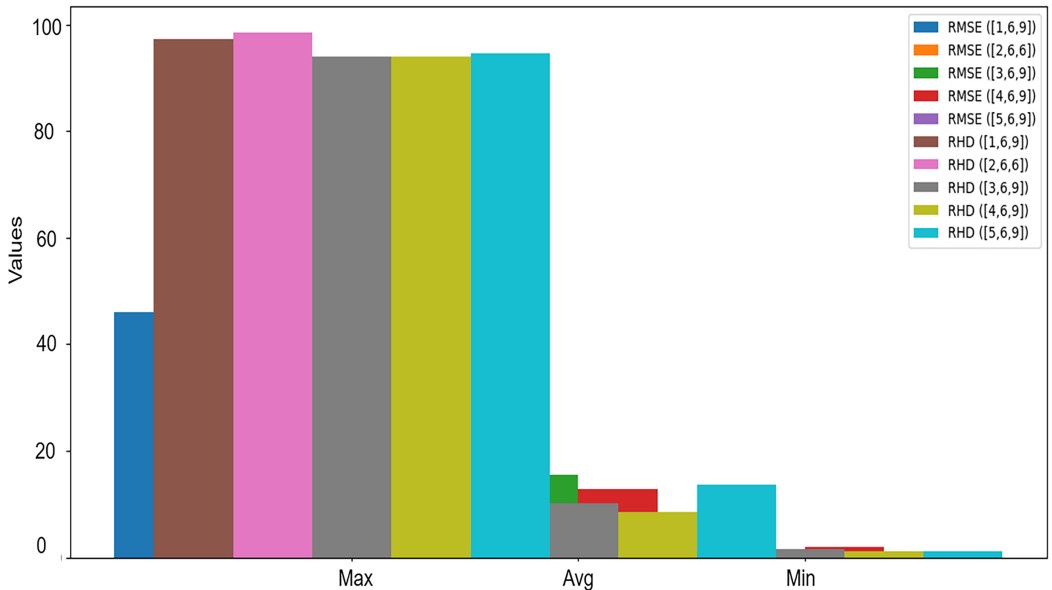

**Figure 9 Comparative analysis of RMSE and RHD statistics.**

**Table 11 Comparison between proposed 3D steganography model and state-of-the-art methods.**

| Ref | Bit selection | Mesh | Vertex shuffling | Random sampling | ROI |
|---|---|---|---|---|---|
| *Wang, Xu & Li (2019)* | No | No | Yes | No | No |
| *Gao et al. (2023)* | Yes | Yes | Yes | Yes | Yes |
| *Kostiuk et al. (2022)* | Yes | No | Yes | No | Yes |
| *Liang, He & Li (2019)* | No | Yes | No | Yes | No |
| *Liang et al. (2023)* | Yes | Yes | No | Yes | No |
| *Lyu, Cheng & Yin (2022)* | No | Yes | No | Yes | No |
| Proposed | Yes | Yes | No | Yes | Yes |

97.29. This represents an essential element of the proposed secure encryption scheme. Compared to other ciphertext 3D models, such as the Statue of Liberty, this 3D model has relatively lower NPCR values than the larger images; however, it retains a satisfactory level of security, as illustrated in Fig. 11. The Bugati and Harpy routinely exhibit elevated NPCR, signifying robust encryption efficacy. Nevertheless, the disparities are negligible and do not substantially affect the overall security. This study successfully makes the encrypted 3D models unexpected, hence hindering the ability of external parties to revert to the original plaintext.

## Adversarial robustness analysis

We conducted a statistical analysis utilizing histograms and descriptive statistics for geometric characteristics, as shown in Fig. 12. We conducted these tests on each 3D model, utilizing both the original cover 3D model and the stego 3D model. This test analyzed vertex coordinates, edge lengths, and face areas. The statistical analysis indicates no
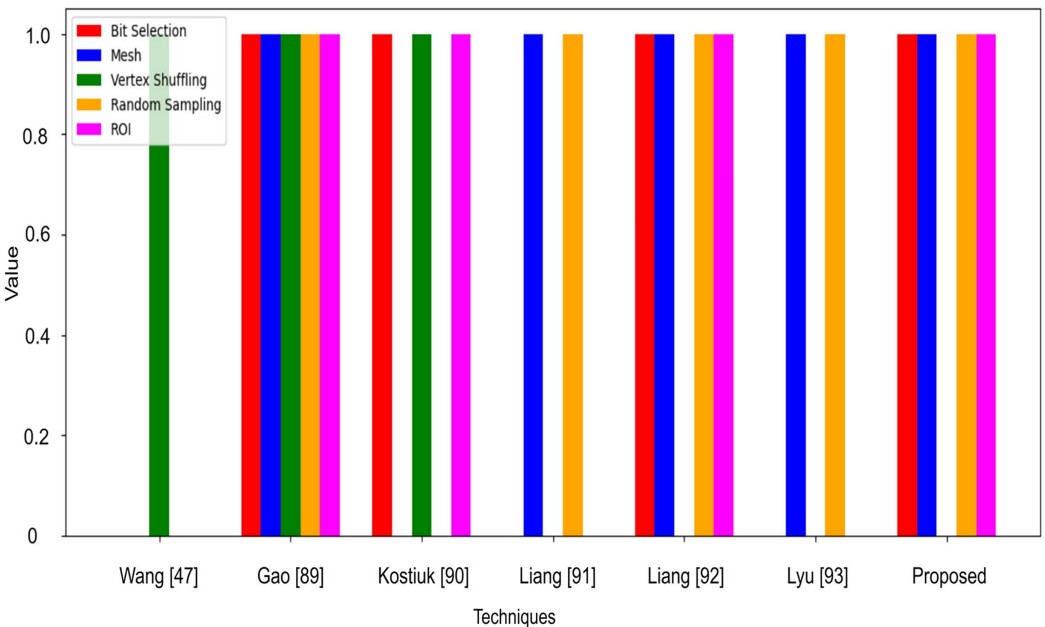

**Figure 10 RMSE and RHD comparison analysis in terms of Min, Avg, and Max values.**

**Table 12 Comparison of NPCR and UACI.**

| 3D model | UACI | NPCR |
|---|---|---|
| Bugati | 99.56 | 32.40 |
| Liberty statue | 98.69 | 31.32 |
| Hand | 99.38 | 34.66 |
| Zombie hand | 97.29 | 32.98 |
| Harpy | 98.54 | 32.49 |

differences between the original cover model and the stego model following the embedding of concealed data. The distributions of vertex X-coordinates demonstrate no statistical difference (KS-test $p$-value = 0.865), suggesting that the steganographic procedure has not altered the spatial arrangement of vertices. Both models have comparable overall geometric configurations. The distributions of edge lengths are highly comparable, exhibiting a minimal KL divergence of 0.0153, indicating that the steganographic embedding successfully maintains local connectivity and edge-related geometric characteristics. The distributions of face areas exhibit the same patterns, signifying that the surface topology remains constant. The results indicate that the steganographic technique effectively embeds data. The maintenance of edge lengths and face areas signifies robust structural integrity, whereas the vertex coordinates imply that the embedding procedure exhibits no visible alterations between the original and the stego 3D models.

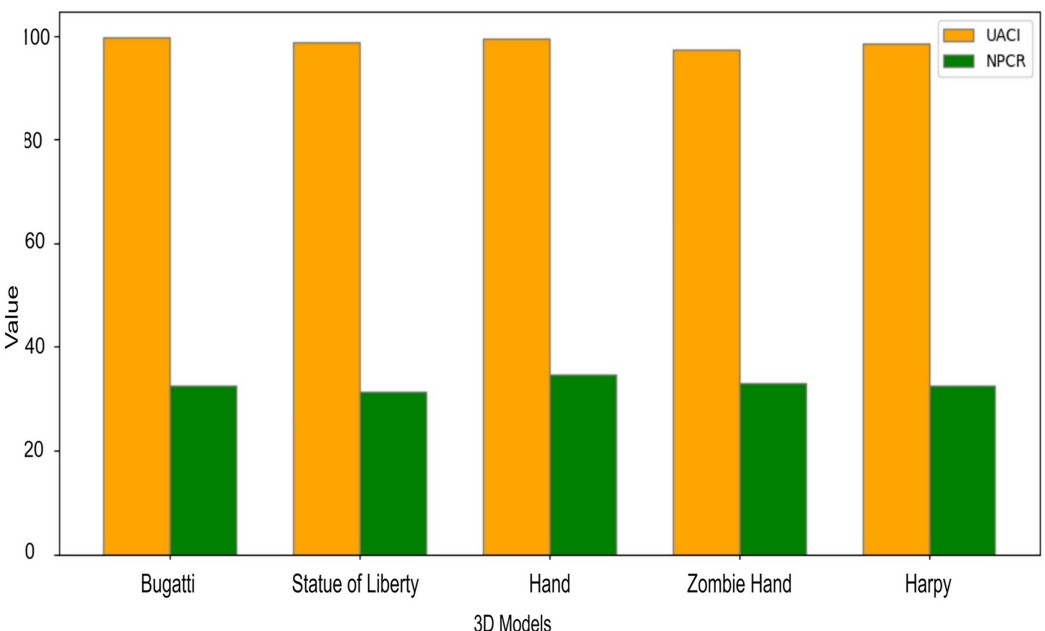

**Figure 11 UACI and NPCR comparison for performance analysis of proposed 3D steganography.**

## Analysis of computational efficiency and optimization strategies and trade-offs

Table 8 shows how efficient the computations are, with insertion times ranging from 0.021 to 0.7913 s and extraction times from 0.0023 to 0.491 s. These indicators elucidate significant attributes of our methodology: Insertion procedures are consistently more computationally demanding than extractions, and processing time scales non-linearly with the complexity of the model. Although rapid operations are appropriate for real-time applications, the upper-bound performance may be excessive for latency-sensitive scenarios like interactive AR/VR environments. To tackle these efficiency difficulties while preserving our security assurances, multiple optimization avenues are available. We can employ dynamic mesh decimation to mitigate processing overhead for intricate models; however, such methods may marginally elevate geometric distortion. Hybrid encryption systems can use AES-128 for important data and faster algorithms like ChaCha20 for less sensitive information, which improves speed while still keeping basic security. The parallel processing of mesh sub-regions may utilize contemporary multi-core architectures to expedite embedding procedures, albeit this entails heightened memory demands. In situations where processing is done in groups, saving commonly used embedding areas can greatly reduce computation time, but this may make it harder to adjust the model later. Each optimization involves specific trade-offs between speed, security, and model integrity, which require meticulous management based on application needs. Future research will statistically assess these solutions using standardized datasets to create definitive performance benchmarks while preserving the strong security and visual quality evidenced in our current findings.

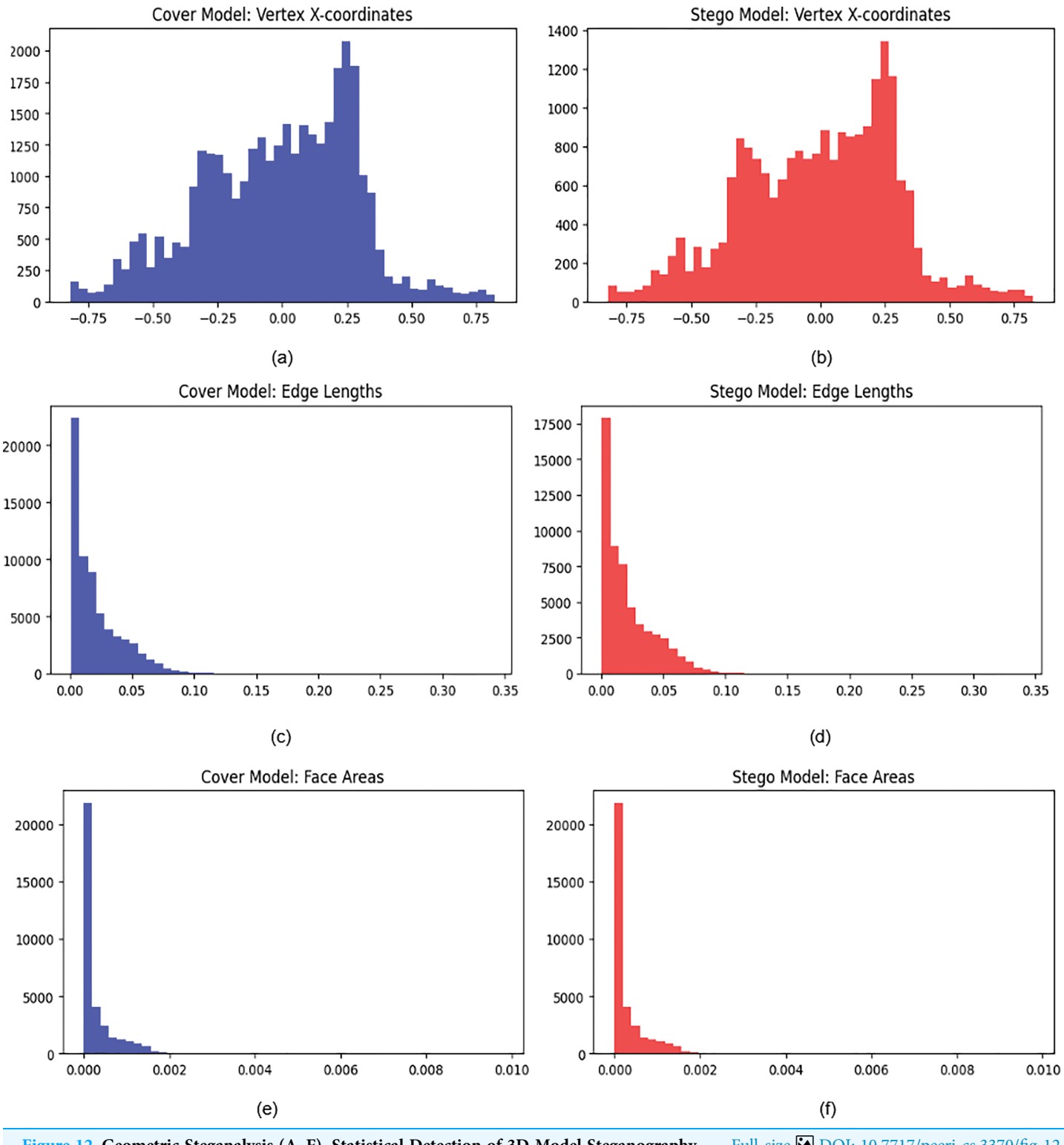

**Figure 12  Geometric Steganalysis (A–F), Statistical Detection of 3D Model Steganography.**

## Security analysis

The dual-layered approach significantly enhances the overall security of hidden data. This study has implemented multiple layers of defense to fortify the proposed model against identified vulnerabilities.

- Brute-Force Attack: The strength of AES-128 encryption makes brute-force attacks computationally unfeasible for most applications. These issues ensure that the proposed architecture not only safeguards data during transmission but also continuously prevents unauthorized decryption efforts. This establishes a robust foundation for transmitting sensitive 3D models.
- Prevention of unintentional assaults: The proposed model's characteristics make it robust against external assaults and internal risks posed by authorized users attempting to manipulate the transmission process. Thus, the proposed method exhibits robustness against assaults and is an essential component of data confidentiality and cybersecurity.
- Geometric transformation: To mitigate geometric attacks, we utilized strong encryption techniques such as AES-128, applied resilient hashing algorithms like SHA-256, chose features of the 3D framework for data that are less susceptible to standard tampering and put into effect adaptive strategies that adjust the embedding method based on the model's characteristics and possible threats.

## Key benefits of the integrated approach

The primary innovation of our proposed technique is not in the creation of wholly new components but in the intricate and intentional amalgamation of existing techniques into a novel, resilient framework specifically tailored to mitigate the distinct weaknesses of 3D model security. This study significantly advances the field by transcending basic encryption and simple steganography; it presents a multi-layered protection system that exceeds the cumulative effectiveness of its components. The technological progress is realized by the strategic integration of AES-128-CBC cryptography with a geometrically aware steganographic algorithm, resulting in a system where each layer compensates for the vulnerabilities of the other. The technique for detecting and employing conspicuous vertices for data concealment, based on inherent geometric characteristics, represents the principal innovative advancement. This methodology fundamentally differs from methods that modify random or less significant vertices, as it deliberately utilizes areas of high geometric significance. This intentional decision improves security against steganalysis by rendering modifications less statistically abnormal while concurrently guaranteeing that the embedded data remains robust to typical mesh processing procedures that may impact less critical regions. The technology addresses the fundamental weaknesses of current 3D data transfer systems by resolving geometric inconsistencies and skillfully masking deformations. This article presents an innovative, comprehensive security framework for 3D models that achieves an optimal equilibrium of imperceptibility, capacity, and robustness, thereby effectively tackling a significant challenge in the secure transfer of assets within the metaverse.

## Limitations and future directions

The proposed 3D Steganography model showed exceptional performance in evaluation criteria and established a robust foundation for safeguarding 3D models. Nonetheless, we recognize the following limitations that we will rectify in the future.

- The present study utilizes custom-generated 3D models with a limited selection of publicly available 3D models. Though the results are promising, the limited selection of 3D models may introduce biases that affect the assessment of the generalizability and reproducibility of our steganographic methodology. To mitigate this restriction, further research will integrate established standards such as ShapeNet and Princeton ModelNet.
- Although our existing methodology exhibits strong security and steganographic efficacy, we recognize significant computing constraints that necessitate additional optimization. These will be methodically handled in the future.
- Our methodology utilizes AES-128 in CBC mode for encryption and SHA-256 for integrity verification, establishing a strong security framework. This combination ensures that data is kept private, and it also guarantees that the data hasn't been changed. In the future, we intend to investigate ChaCha20 for mobile and edge implementations and assess Keccak (SHA-3) for post-quantum preparedness, ensuring our architecture remains responsive to changing security requirements and hardware environments.

## CONCLUSION

The emergence of the metaverse has sparked significant interest in 3D models, although data transfer security remains a critical concern. In the modern digital environment, marked by pervasive internet access and extensive image dissemination, safeguarding sensitive information within 3D models is becoming increasingly essential. This work presents a modern and efficient system that integrates cryptography with 3D steganography techniques. This study utilized AES-128 with cipher block chaining (CBC-IV) and an initialization vector to transform plaintext into ciphertext. The research utilized SHA-256, salt, and a 32-bit password to generate the encryption key, establishing a basic layer of security. This study utilized encrypted data in a 3D facial model, leveraging geometric attributes. This study delineated critical regions, selected notable vertices, and evaluated the significance of each vertex based on geometric attributes. The current work incorporated data into vertices next to landmarks by rounding and an enhanced scale factor, resulting in a stego 3D model. The results showed that the proposed model performed very well, achieving a PSNR of 61.31 dB, an MSE of 3.17, a correlation coefficient of 0.95, and a Harsdorf distance of 0.04. This study attained notable NPCR and UACI ratings of 94.82 and 28.31, respectively, when compared to other established methodologies. Our methodology resolves geometric inconsistency issues and skillfully obscures the model's distorted geometry.

In the future, we intend to investigate the proposed 3D steganography technique across many types of 3D models, including architectural, cultural, and medical 3D models. We also intend to utilize our concept for applications in intellectual property protection, authentication, and clandestine communication within the metaverse. We aim to integrate

this methodology with blockchain technology to improve security, immutability, and provenance tracking for digital assets. In the future revisions, we also intend to concentrate on modifying the methodology for dynamic and animated 3D objects, ensuring resilience under diverse rendering and transmission settings, thereby expanding the domain of secure and reliable 3D data handling.

### Funding
The authors received no funding for this work.

### Competing Interests
The authors declare that they have no competing interests.

### Author Contributions
- Muhammad Sajid conceived and designed the experiments, performed the experiments, performed the computation work, prepared figures and/or tables, and approved the final draft.
- Kaleem Razzaq Malik conceived and designed the experiments, performed the experiments, performed the computation work, prepared figures and/or tables, and approved the final draft.
- Sohail Jabbar analyzed the data, authored or reviewed drafts of the article, and approved the final draft.
- Umar Raza analyzed the data, authored or reviewed drafts of the article, and approved the final draft.
- Muhammad Asif Habib performed the computation work, authored or reviewed drafts of the article, and approved the final draft.

### Data Availability
The raw data is available at Zenodo: Sajid, M. (2025). 3D Models Dataset. Zenodo. https://doi.org/10.5281/zenodo.15299033. All data supporting this study are openly available within the article and online as open source (*Free3D, 2025*), without restrictions.

The code is available at Zenodo: Muhammad Sajid. (2025). auksajid/3D-Model-Steganography: Securing 3D Models (V2.0). Zenodo. https://doi.org/10.5281/zenodo.15299260. The implemented code used in this study can be found at (*Khan & Malik, 2025*), which contains code configuration files, source code, and a user manual.

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
