# Peer review of "Secure 3D data hiding through cryptographic steganalysis resistance: reducing geometric inconsistency vulnerabilities"

_PeerJ Computer Science, doi:10.7717/peerj-cs.3370_

## Round 0.1 · original submission · Major Revisions

· Academic Editor

Major Revisions

• I think it would be a nice paper if it were clearly organized and the main conclusion was clearly supported by the analysis and figure interpretations.

• I think that the manuscript has to be significantly revised for clarity, and there are some missing points that the authors have to rewrite. The methods are appropriate but not well described in sufficient detail.

• The manuscript could be original research if the text were well described and organized, with a good reason for such studies.

• The authors have to look ahead through their abstract. They have to clarify what problem they want to solve and what their findings are.

• They have to reduce the keywords to 5.

• The authors should start the introduction with a brief history of their analysis and method. I suggest that they consider both advantages and inconveniences before stating what they are look- ing for, with the limit found in the literature. Also, we do not understand which problem they want to solve. Can they give more explanation? There are some grammatical errors in the text; they have to be checked. They also have to provide more applications concerning their studies and methods. There are some repeating sentences at the beginning of the introduction.

• What relevance does the proposed encryption method have compared to existing ones in the literature? Are they the first to work on it? How do they measure the robustness of their secured communication method?.

• They have to provide a good introductory part to the section concerning the encryption. Which algorithms have they used? Provide their limits and their advantages compared to the others found in the literature.

• The in-text citations should be linked to the bibliography list at the end of the manuscript. Also, no more than three citations should be inserted for a single statement. The extra ones should be removed or moved to other suitable places.

• The in-text equation, figure, section, and table numbers should be linked to their corresponding position in the manuscript.

• The results of the Figures should be discussed in depth and detail. What is their message, and what makes the authors report them?

Reviewer 1 ·

Basic reporting

-

Experimental design

-

Validity of the findings

-

Additional comments

The metaverse’s rise has sparked interest in 3D models, but data security remains critical. This paper presents a system combining cryptography and 3D steganography. Using AES-128 with CBC-IV for encryption and SHA-256 for key generation, sensitive data is protected within 3D facial models by focusing on significant geometric vertices. The study achieved strong performance metrics: PSNR of 61.31 dB, MSE of 3.17, a correlation coefficient of 0.95, and a Hausdorff distance of 0.04. High NPCR and UACI scores of 94.82 and 28.31 were also noted.

General remarks and suggestions:

• I think it would be a nice paper if it were clearly organized and the main conclusion was clearly supported by the analysis and figure interpretations.

• I think that the manuscript has to be significantly revised for clarity, and there are some missing points that the authors have to rewrite. The methods are appropriate but not well described in sufficient detail.

• The manuscript could be original research if the text were well described and organized, with a good reason for such studies.

• The authors have to look ahead through their abstract. They have to clarify what problem they want to solve and what their findings are.

• They have to reduce the keywords to 5.

• The authors should start the introduction with a brief history of their analysis and method. I suggest that they consider both advantages and inconveniences before stating what they are look- ing for, with the limit found in the literature. Also, we do not understand which problem they want to solve. Can they give more explanation? There are some grammatical errors in the text; they have to be checked. They also have to provide more applications concerning their studies and methods. There are some repeating sentences at the beginning of the introduction.

• What relevance does the proposed encryption method have compared to existing ones in the literature? Are they the first to work on it? How do they measure the robustness of their secured communication method?.

• They have to provide a good introductory part to the section concerning the encryption. Which algorithms have they used? Provide their limits and their advantages compared to the others found in the literature.

• The in-text citations should be linked to the bibliography list at the end of the manuscript. Also, no more than three citations should be inserted for a single statement. The extra ones should be removed or moved to other suitable places.

• The in-text equation, figure, section, and table numbers should be linked to their corresponding position in the manuscript.

• The results of the Figures should be discussed in depth and detail. What is their message, and what makes the authors report them?

• The manuscript should include ‘Future works’ and ‘Authors' contributions’ sections. Once these questions are addressed, the manuscript should be published.

Reviewer 2 ·

Basic reporting

The advent of the metaverse has generated considerable interest in 3D models, although data transfer security continues to be a paramount issue. In the contemporary digital landscape, characterized by ubiquitous internet connectivity and widespread image distribution, the protection of sensitive data within 3D models is becoming increasingly imperative. To this end, this manuscript presents a contemporary and effective system that combines cryptography with 3D steganography techniques.

Experimental design

For experimental details, the authors present the implementation details and evaluate the performance of the system, which are interesting and convincing. According to “section RESULTS AND DISCUSSION” and “section Dataset”, this manuscript introduces an integrated method for 3D models in the OBJ file format. However, it is not clear whether the dataset is publicly available. Thus, it would be better to use more recent public datasets for experiments. This comment is optional but not necessary.

Validity of the findings

The proposed method is reasonable.

This manuscript employed AES-128 with cipher block chaining (CBC-IV) and an initialization vector to convert plaintext into ciphertext. The study employed SHA-256, salt, and a 32-bit password to produce the encryption key, creating a fundamental layer of protection. This research used encrypted data within a 3D facial model employing geometric characteristics. This study defined key regions, identified significant vertices, and assessed the importance of each vertex based on geometric characteristics. The present study included data on vertices adjacent to landmarks, which were rounded and augmented using an enlarged scale factor, resulting in a stego 3D model. Thus, the proposed method is reasonable.

Additional comments

(1) The authors must further emphasize the novelty of the proposed method, particularly in the theoretical framework, to clearly articulate its contributions. The manuscript should highlight how this work skillfully integrates multiple concepts and techniques, such as steganalysis, salient vertices, vulnerabilities, data hiding, 3D models, and geometric analysis. This combination represents a blend of novelty and incremental innovation, which must be explicitly stated to strengthen the case for the method’s originality and value.

(2) To demonstrate the broader applicability of the proposed idea, the authors can discuss potential extensions to other tasks in the “Conclusion and Future Work” section.

(3) To enhance reproducibility and support the research community, the authors are strongly encouraged to make their resources publicly available; this includes the dataset, algorithm source code, and data preparation scripts, preferably via a platform like GitHub. While optional, open-sourcing these materials would significantly increase the impact and credibility of the work.

(4) The manuscript’s readability and overall presentation must be improved. Figures and tables should be enhanced for better clarity and visual impact. Additionally, the authors must carefully review the manuscript for language accuracy, correct use of coefficients, and consistent functional notation throughout.

Reviewer 3 ·

Basic reporting

• Strengths:
The manuscript provides a thorough and technically rich background.
Figures and tables are clearly labeled and relevant.
A large number of references (75+) are used, showing extensive literature coverage.

• Weaknesses:
The author does not clearly identify the innovative contributions of this paper. Moreover, the connection between the proposed innovations and the stated motivation is unclear. For example, in the abstract, the author mentions the use of AES-128 with CBC-IV and the use of SHA-256 with salt and a 32-bit password to generate the encryption key. However, the relationship between these two technologies is not clearly explained.

There are typos in Algorithm 1 and Algorithm 2.
English grammar and clarity need improvement. The paper often uses overly complex or awkward constructions.
The writing is verbose and repetitive in many areas, reducing readability.

• Suggestions:
Conduct a professional language edit to improve clarity and tone.
Streamline and shorten sections, especially the Abstract and Introduction.
Make the abstract more reader-friendly by briefly describing the innovation and results at a high level.

Experimental design

Strengths:
The problem—protecting 3D model data—is relevant and underexplored.
A clear methodological pipeline is presented: AES-128 encryption, landmark-based vertex embedding, and performance evaluation via multiple metrics.

Weaknesses:
The experimental procedure is described in overwhelming mathematical detail without a high-level summary. It’s difficult for readers to grasp the flow without scanning multiple pages.
Some crucial design decisions are not justified, e.g., why particular vertices are selected, or how embedding thresholds are tuned.
The encryption techniques (AES, SHA-256) are standard. The novelty lies in integration with 3D steganography, but this contribution could be clearer.
The experimental results are weak. The author conducted experiments on only five 3D models, which lack diversity. Additionally, only 2D image files are provided in the supplemental materials.

Validity of the findings

The 3D models referenced in the paper are not provided. Only 2D images are included in the supplementary materials.

In addition, the experimental results are too limited to support the claims made in the paper.

---

## Round 0.2 · Minor Revisions

· Academic Editor

Minor Revisions

Please address the requests and comments thoroughly. One reviewer has specific requests.

Reviewer 1 ·

Basic reporting

Accept.

Experimental design

Good.

Validity of the findings

Good.

Additional comments

No.

Reviewer 2 ·

Basic reporting

(1) Motivation and summary

The advent of the metaverse has generated considerable interest in 3D models, although data transfer security continues to be a paramount issue. In the contemporary digital landscape, characterized by ubiquitous internet connectivity and widespread image distribution, the protection of sensitive data within 3D models is becoming increasingly imperative. To this end, this manuscript presents contemporary and effective system that combines cryptography with 3D steganography techniques.

Experimental design

(2) The proposed method is reasonable

This manuscript employed AES-128 with cipher block chaining (CBC-IV) and an initialization vector to convert plaintext into ciphertext. The study employed SHA-256, salt, and a 32-bit password to produce the encryption key, creating a fundamental layer of protection. This research used encrypted data within a 3D facial model employing geometric characteristics. This study defined key regions, identified significant vertices, and assessed the importance of each vertex based on geometric characteristics. The present study included data on vertices adjacent to landmarks, which were rounded and augmented using an enlarged scale factor, resulting in a stego 3D model. Thus, the proposed method is reasonable.

Validity of the findings

(3) Advantages

Overall, the topic addressed in the paper is potentially interesting. The motivation of the article is clear. The overall scheme of the manuscript is reasonable. The proposed algorithm pipeline and mathematical developments put forward some novelty in the topic. The findings indicated that the proposed model displayed strong performance, with a PSNR of 61.31 dB, an MSE of 3.17, a correlation coefficient of 0.95, and a Hausdorû distance of 0.04. This study achieved significant NPCR and UACI scores of 94.82 and 28.31, respectively, in comparison to other recognized methods. The proposed methodology addresses geometric inconsistency issues and adeptly conceals the model9s deformed geometry.

Additional comments

(4) For technical novelty, the authors are suggested to further emphasize new or novelty of this proposed method through theory part, in order to show the contributions of this method. For example, this work skillfully combines several concepts, approaches, techniques and components, such as: Steganalysis, Salient Vertices, Vulnerabilities, Data Hiding, 3D Models, Geometric. It is a typical combination novelty and/or increment novelty, which can be further highlighted to show the contributions and/or advantages of the proposed method.

(5) For generality exploration, in order to show the beauty of the proposed idea, it would be better to discuss how to extend the proposed idea to other tasks in section of “Conclusion and Future Work”.

(6) For reproducibility, it would be better to open the source (data source and/or algorithm source code and/or data preparing code) on GitHub. The open source will greatly contribute to the community of related research areas. This comment is optional but not necessary.

(7) The readability and presentation of this manuscript can be improved. Visualization of some figures and tables can be enhanced. Please carefully read and check the language, coefficients, functional, notations throughout the manuscript.

---

## Round 0.3 · accepted · Accept

· Academic Editor

Accept

Thank you for your valuable contribution.

Reviewer 1 ·

Basic reporting

Accept.

Experimental design

Accept.

Validity of the findings

Accept.

Additional comments

Accept.